# Bioactive Compounds from Vegetal Organs of *Taraxacum* Species (Dandelion) with Biomedical Applications: A Review

**DOI:** 10.3390/ijms26020450

**Published:** 2025-01-07

**Authors:** Maria-Virginia Tanasa (Acretei), Ticuta Negreanu-Pirjol, Laura Olariu, Bogdan-Stefan Negreanu-Pirjol, Anca-Cristina Lepadatu, Larisa Anghel (Cireasa), Natalia Rosoiu

**Affiliations:** 1Institute of Doctoral Studies, Doctoral School of Applied Sciences, Doctoral Field:Biology, “Ovidius” University of Constanta, 58, Ion Voda Street, 900573 Constanta, Romania; tmariavirginia@yahoo.com (M.-V.T.); cireasa.larisa-ct@ansvsa.ro (L.A.); natalia_rosoiu@yahoo.com (N.R.); 2Faculty of Pharmacy, “Ovidius” University of Constanta, 6, Capitan Aviator Al. Serbanescu Street, Campus, Building C, 900470 Constanta, Romania; 3Academy of Romanian Scientists, Biological Sciences Section, 3, Ilfov Street, 50044 Bucharest, Romania; olariulaura@yahoo.com; 4S.C. Biotehnos S.A., Gorunului Street, No. 3-5, Ilfov County, 075100 Bucharest, Romania; 5Faculty of Natural Sciences and Agricultural Sciences, “Ovidius” University of Constanta, 1, University Alley, Campus, Building B, 900470 Constanta, Romania; anca.lepadatu@365.univ-ovidius.ro

**Keywords:** *Taraxacum officinale*, phytotherapeutic plant, antioxidant, antimicrobial, anticancer, anti-inflammatory, hepatoprotective and antidiabetic properties

## Abstract

*Taraxacum officinale* (dandelion) is a perennial flowering plant of the Asteraceae family that has spread globally and is well-known for its traditional uses. The aim of this work is to provide a detailed review of scientific literature on the genus *Taraxacum* from the last two decades, with particular emphasis on the biological and pharmacological characteristics of dandelions. The traditional use of *Taraxacum* species and their potential use in medicine are assessed. In addition, individual papers describing principal pathways and molecules modulated by *Taraxacum* in antitumoral, anti-inflammatory, antidiabetic, hepatoprotective, immunomodulatory, antimicrobial, and antioxidant activities are presented. This review of phytochemical studies reveals that dandelions contain a wide range of bioactive compounds, such as polyphenols, phytosterols, flavonoids, carotenoids, terpene, and coumarins, whose biological activities are actively explored in various areas of human health, some constituents having synergistic activities, including antioxidant, antimicrobial, anti-inflammatory and anticancer activities. The study provides a screening of *Taraxacum* sp. chemical composition, an assessment of the main pharmacological properties, and a description of relevant studies supporting the use of dandelion for its particularly valuable and diversified therapeutic potential in different diseases.

## 1. Introduction

Phyto-therapeutic plants have always played a significant role in folk medicine in remedying a wide range of medical problems. Approximately 80% of the global population relies on plant-based remedies to treat various ailments, according to the World Health Organization [1]. As society progresses and technology advances, plants have emerged as a valuable therapeutic resource. They contain biologically active compounds that can be used to synthesize numerous synthetic substances for treating human diseases.

Within the biomedical field, phytotherapeutic drugs offer a promising alternative. These drugs are derived from plants, making them safer, more sustainable, and less harmful to the environment. They are also less likely to lead to bacterial resistance or leave behind persistent chemical residues while still effectively addressing therapeutic needs.

Plants and plant-based products have shown considerable promise in treating inflammatory conditions and managing diseases with significant inflammatory components. Many medicinal plants used as anti-inflammatory agents offer advantages over conventional non-steroidal anti-inflammatory drugs (NSAIDs), as they often lack the gastrointestinal side effects and other unwanted adverse reactions associated with NSAIDs. Additionally, there is a growing trend towards using natural food additives due to their accessibility and perceived safety. Phenolic compounds, when incorporated into fresh, biodegradable foods, have shown potential as natural preservatives, offering a viable alternative to synthetic food additives [2]. Methods like dipping, spraying, and coating are currently employed in food technology to treat food before packaging [3].

Asteraceae, one of the largest among flowering plants, encompasses nearly 1620 different types and over 23,000 species. These plants are remarkably adaptable, thriving in diverse climates and habitats worldwide. The family is divided into three major subfamilies: Asteroideae, Barnadesioideae, and Cichorioideae. Asteraceae plants exhibit a wide range of forms, including herbs, shrubs, and trees. They are easily recognized by their composite flower heads and single-seeded fruits.

Asteraceae species produce a variety of compounds with biological activities that mirror those of the plants themselves. Many of these species are integral to human diets, while others have found applications in medicine, cosmetics, and food industries. For centuries, *Taraxacum officinale*, or dandelion, has been used in traditional medicine to treat liver, kidney, lung disorders, and diabetes. This is attributed to its anti-inflammatory, antimicrobial, antioxidant, and immune-boosting properties.

It has spread worldwide and is widely used in a variety of foods and nutraceuticals. The plant’s leaves are consumed as a leafy green vegetable. Its bitter leaves serve as a component in various beverages, including wines, beers, and soft drinks. The ground roots can be used as a coffee alternative. The unopened flower buds can be utilized as a culinary substitute for capers. In certain regions, the plant has historically been employed as a nutritional supplement but also as an herbal remedy for the prevention, management, and treatment of various human diseases.

*Taraxacum* species are remarkable plant due to their phytochemical combination: polyphenols, flavonoids, phytosterols and sesquiterpenoids, inositol derivates, polysaccharides, carotenoids, coumarins, and mineral elements. Polyphenols are recognized as being an important class of *Taraxacum* extract with antiviral effects and protect the body from oxidative stress, diabetes, cardiovascular diseases, and cancer. Flavonoids are known as relevant for antioxidant, antiviral, anti-mutagenic, anti-inflammatory, and cardioprotective roles; sesquiterpenoids and sterols are remarkable for their anticancer and anti-inflammatory activity, and polysaccharides are significant for their anti-inflammatory, antioxidant and antidiabetic actions.

To uncover recent insights into the last decade’s research on *Taraxacum*, a comprehensive literature review was conducted using multiple databases. A systematic search process was employed, ensuring articles focused on all aspects of *Taraxacum*, including botany, traditional uses, phytochemistry, and pharmacology. To ensure the relevance of the included studies, we established rigorous inclusion and exclusion criteria. After an initial title and abstract screening, full-text articles were carefully evaluated to determine their suitability for in-depth analysis. This review provides a comprehensive overview of recent research on *Taraxacum*, with a particular focus on its traditional uses and pharmacological potential. Various electronic databases, including ScienceDirect, Elsevier, Scopus, and Google Scholar, and keywords such as *“Taraxacum*”, “medicinal plant”, “phytochemistry”, “biomedical effects”, etc., were used to identify relevant publications from the year 2000 to 2024.

The search on the Elsevier database found 662 articles using “*Taraxacum*, medicinal plant” keywords, 661 for “*Taraxacum*, antioxidant activity”, 414 for “*Taraxacum*, antimicrobial activity”, 323 for “*Taraxacum*, biomedical effects”, 282 for “*Taraxacum*, anticancer activity”, 166 for “*Taraxacum*, immunomodulatory effect” and 79 for the “*Taraxacum*, phytochemistry” keywords. The search on Google Academic database revealed 12,300 articles for “*Taraxacum*, medicinal plant” keywords, 8140 for “*Taraxacum*, antioxidant activity”, and 5930 for the “*Taraxacum*, phytochemistry”, 5070 for “*Taraxacum*, antimicrobial activity”, 3600 for “*Taraxacum*, anticancer activity”, 2150 for “*Taraxacum*, biomedical effects”, 1740 for “*Taraxacum*, immunomodulatory effect” keywords. Search terms were run separately or in limited combinations according to the requirements or limitations of the database used. A systematic literature search was undertaken to identify articles published between 2000 and 2024. After excluding irrelevant studies, a total of 310 articles were selected for detailed review (Figure 1). The objective of this review is to emphasize the significance of research focused on the common dandelion (*Taraxacum*) with a particular focus on its botanical, phytochemical, and pharmacological properties.

## 2. General Description of the Species

*Taraxacum officinale* (L.) Weber ex F.H.Wigg species, commonly known as the dandelion, is a resilient herb that can thrive in diverse environments. Found on every continent except Antarctica [4], this versatile plant can grow from sea level to alpine zones and adapt to a wide range of soil types. The Swedish botanist Carl Linnaeus named the species *Leontodon taraxacum* in 1753 [5], but Wiggers (1746—1811) described the genus *Taraxacum*, and the German botanist Georg H. Weber named the plant *Taraxacum officinale* in 1780 [6]. The genus name *Taraxacum* is derived from the Greek word “taraxos”, meaning “agitation” or “disorder”, and “akos”, meaning “remedy” [7,8]; *officinale* means medicinal or capable of producing medicine [9], or “of the shops”, meaning it was sold as a remedy for man’s illnesses [10]. The first mention of dandelion as a remedy was found in the works of Arab doctors in the tenth and eleventh centuries, who speak of it as *Taraxacum*.

The dandelion root is primarily recognized for its digestive and liver-supporting properties [11,12,13], while the leaves are used as a diuretic and digestive stimulant. Preclinical research has uncovered various pharmacological activities of dandelion extracts, including anti-inflammatory, diuretic, digestive stimulant, insulin-stimulating, anti-angiogenic, and anti-neoplastic effects.

Dandelion contains a diverse array of bioactive compounds, such as terpene derivatives (taraxerol, taraxasterol, taraxacoside) [14], sterols (β-sitosterol, stigmasterol), phenolic compounds (hydroxycinnamic acids, coumarins, lignans) [15], rubber, resins [16], tannins, fatty acids, sugars, amino acids, carotenoids, phytosterols, and inulin [17,18,19].

### 2.1. Taraxacum Plant Characterization

*Taraxacum officinale* (L.) Weber ex F.H.Wigg is a perennial laticiferous plant with a milky latex, up to 30–50 cm tall. It has a long root, 15 to 50 cm long and 1.5–2.5 cm in diameter. The plant boasts a thick, branched taproot that is dark grey externally and white pale internally, with lateral roots wind in a loose spiral around the main root and distributed along its length (Figure 2A).

*Taraxacum* possesses a short stem with subterranean internodes. The basal rosette of leaves [20], which can range from horizontal to nearly vertical, exhibits significant morphological variability. The leaves (Figure 2B), often with winged petioles, can be entire or deeply lobed, with the margins ranging from smooth to deep toothed. Each stalk supports a solitary, terminal flower head composed of numerous yellow-orange florets (Figure 2C). The fruit is a brownish oblong cypsela equipped with a white pappus for wind dispersal [21].

Before flowering, cylindrical stalks emerge from the center of the leaf rosette. Each stalk supports a single terminal flower head composed of numerous yellow-orange florets. Each perfect flower has a fused petal tube with a strap-shaped extension and five fused stamens with a distinctive base. The fruit is a brownish oblong cypsela, a few millimeters long and with a grooved surface, a cotton-like ball with numerous seeds (Figure 2D). A white pappus atop each cypsela allows the wind to disperse it because the white pappus has numerous hairs, mostly white and persistent [20]. Connecting region of a dandelion seed includes barbs distributed on the inserting region of achene and the groove of the inflorescence head. The barbs have conical morphology, and the groove has plenty of wrinkles to enhance its wrapping ability [21].

### 2.2. Phytochemistry

Advanced techniques are necessary for the precise identification and quantification of phytochemicals. High-performance liquid chromatography (HPLC), liquid chromatography-electrospray ionization-mass spectrometry (LC-ESI-MS), ultra-high-performance liquid chromatography-quadrupole time-of-flight mass spectrometry (UHPLC/HR-QTOF-MS), and nuclear magnetic resonance (NMR) spectroscopy are commonly employed for the sensitive analysis of these compounds [22,23,24,25,26]. For instance, HPLC has been used to identify luteolin, luteolin-7-O-glucoside, caffeic acid, and chlorogenic acid in various fractions of plants [27]. In other studies, two novel guaianolide glucosides, deacetylmaricarin 8-O-β-glucopyranoside, and 11β-hydroxyleukodin 11-O-β-glucopyranoside, were isolated from *Taraxacum obovatum* [28]. Additionally, taraxafolide and taraxafolin-B (shown in Table 1) were identified in *Taraxacum formosanum* [29]. The Key HMBC correlations of compounds 1, 2, and 3 are illustrated in Figure 3.

In a flower extract targeted isolation, Jedrzejek and Pawelec, in 2024 [22] revealed three new flavonoids: two biflavones, structured of two luteolin molecules, linked by a C–C bond through C-6′ and C-6″ or C-8″ and novel flavonolignan, composed of tricin and a carboxyl containing a lignan moiety. An increased value of the flavonoids in *T. officinale* was registered by Tanasa et al., 149,864.97 ± 587.45 mg/kg dry weight, for 70:30 (*v:v*) flowers ethanol extract and pretty close to flavonoids concentration values of leaves hydroalcoholic extracts: 138,595.44 ± 755.23 mg/kg dry weight in herba 50:50 (*v:v*) and 135,062.29 ± 881.30 mg/kg dry weight in herba 70:30 (*v:v*) for plants collected from Neamt County, Romania [31]. The influence of climate and soil on *Taraxacum* plants was demonstrated in a comparative phytochemical analysis of root, leaf, and flower extracts in ethyl alcohol by UV-Vis spectrophotometry method. In the warmer region with soil rich in humus and calcium, a higher concentration of carotenoids was found in the root extracts and a higher concentration of lutein and beta-carotene in leaves. In the colder region and with an argillaceous soil, poor in calcium, a higher concentration of lutein was found in root and flower extracts (*T. officinale*, collected in two different Romanian regions: Dobrogea and Harghita County) [32].

In 2022, Dedić et al. [33] analyzed hydroalcoholic extracts obtained from dandelion (*T. officinale*, collected in Bosnia and Herzegovina) by maceration extraction techniques at room temperature for 240 min and boiling temperature—boiling extraction for 30 min, as ultrasonic extraction (Wise Clean WUC) for 30 min and filtered on a Büchner funnel and transferred to vials and Soxhlet extraction, in a Soxhlet apparatus with 25 g of plant material each. The extraction lasted 160 min. The study revealed total phenols in maceration aqueous ethanol solution 50% (*v:v*)—root: 4.23 ± 0.43 mg GAE/g dry matter, leaves: 30.05 ± 0.89 mg GAE/g dry matter and flowers: 26.08 ± 0.67, and flavonoids—in leaves extract 2.26 ± 0.31 mg quercetin/g dry matter and flower 4.98 ± 0.51 mg quercetin/g dry matter. Polyphenols, flavonoids, and carotenoids from the ethanolic extracts of the roots, leaves, and the whole plant were analyzed in a study in 2022, which demonstrated a higher content of polyphenols (possible biosynthetic pathway in Figure 4) and carotenoids in the extract from the leaves and whole plant obtained by cold maceration in 70% ethyl alcohol, and for flavonoids a higher content in the root and leaf extracts at the same concentration of ethyl alcohol [34].

Grauso et al. [35] conducted a detailed analysis of dandelion leaves, identifying several novel compounds in addition to previously known metabolites. Their study highlighted the presence of tartaric acid, formic acid, and sucrose for the first time. Furthermore, caffeoylquinic acid derivatives were found to be the most abundant secondary metabolites, followed by flavonoids. These compounds have been linked to various health benefits, including antioxidant and anti-inflammatory properties. Guo et al. [36] focused on the waxes covering various parts of the dandelion plant. Their research revealed a diverse array of compounds, including novel constituents like C25 and C27 ketols from petal wax. Additionally, they identified nine other classes of compounds: fatty acids, alcohols, aldehydes, esters, alkanes, triterpenes, sterols, and tocopherols. These compounds have potential applications in various industries, such as cosmetics and food.

**Figure 4 ijms-26-00450-f004:**
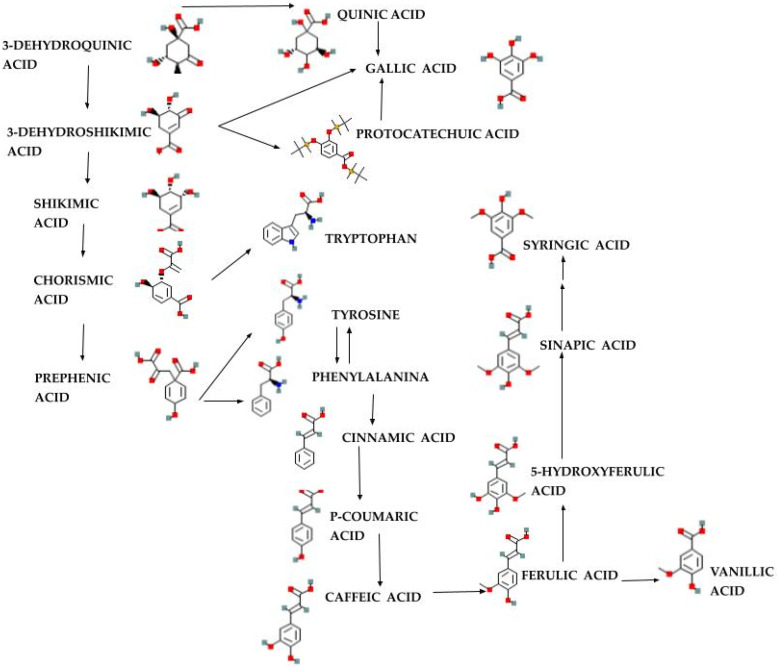
Biosynthetic pathway of some important polyphenols in plants (adapted after [37]).

Esatbeyoglu et al. analyzed dandelion leaves and roots by HPLC/MS to determine their sesquiterpene lactone composition. The main compound isolated from the leaves was taraxinic acid ß-D-glucopyranosyl ester, a sesquiterpene lactone whose structure has been obtained by NMR spectroscopic analysis [38]. Dandelion stems and leaves also possess anthocyanins, carotenoids, and chlorophyll [39]. Three butyrolactones and six butanoates (taraxiroside A-F) were isolated from a methanolic extract of *T. officinale* by Choi et al. in 2018 [40]. Taraxiroside A-F showed α-glucosidase inhibitory activities (IC_50_ 145.3–181.3 μM) similar to that of acarbose. Major sesquiterpene lactones are generally occurring as glycosides—taraxacoides, dihydro-lactucin, ixerin (see Figure 5), and taraxinic acids [41]. Kenny et al., in 2014, found the highest phenolic content (228.723 ± 2.392 mg GAE/g) in the ethyl acetate crude extract of *T. officinale*—root [26]. The bioactive compounds isolated from vegetal organs of *Taraxacum* sp. are described in Table 2.

Mir et al. [42] conducted a comprehensive analysis of bioactive metabolites in different parts of the plant in aqueous and methanol extracts. They found that roots and flowers were particularly rich in saponins, flavonoids, phenols, and alkaloids. Ghaima et al. [43] and Ivanov [44] reported varying levels of total phenols in dandelion leaves; total phenols in dandelion leaves extracted with ethyl acetate was 10.2 mg GAE/g dry matter and respectively 33.90 ± 0.57 mg GAE/g dry matter in 50% ethanol extract highlighting the potential of this plant as a source of antioxidants. Numerous studies have confirmed the presence of essential vitamins, minerals, and other nutrients in dandelion, such as vitamins A, C, D, E, and B, choline, inositol, lecithin, calcium, sodium, magnesium, iron, silicon, copper, phosphorus, zinc, and manganese [45,46].

The objective of the study performed by Popescu et al. in 2010 [47] was the comparative pharmacognostic analysis of the root, stem, leaf, and flower of common dandelion. The chemical analysis determined the presence of flavonoids (rutin, hyperoside, quercetin), hydroxycinnamic acid derivatives (caffeic acid, chlorogenic acid), catechuic tannins, sterols, triterpenes, carotenoids, coumarins, and mucilage. It was observed that flowers and leaves have higher amounts of polyphenols compared to stems and roots. Dandelion petals were examined using thin-layer chromatography (TLC) the name HPLCto investigate the carotenoid lutein epoxide. Six geometric forms were separated and identified based on their UV/VIS spectroscopic characteristics. All-E-lutein epoxide was determined to be the major carotenoid, with substantial quantities of its (9Z)- and (9′Z)-forms [48].

**Table 2 ijms-26-00450-t002:** Bioactive compounds isolated from *Taraxacum* sp. vegetal organs.

Vegetable Organs	*Taraxacum* (T.) Species	Bioactive Compounds	Chemical Formula	Diseases Involved	References
Root	*Taraxacum officinale*	ainsloside	C_37_H_6_2O_16_	antitumor, antioxidant	[49]
		beta-carotene	C_40_H_56_	antitumor, antioxidant, immunostimulant	[32]
		caffeic acid	C_9_H_8_O_4_	anti-inflammatory, antitumor, antioxidant	[50,51]
		caffeoylquinic acid	C_16_H_18_O_9_	anti-influenza	[52]
		chicoric acid	C_22_H_18_O_12_	antioxidant, anti-inflammatory	[53]
		cyclo-artenol	C_30_H_50_O	anti-inflammatory, antitumor, antioxidant	[54]
		faradiol	C_30_H_50_O_2_	anti-inflammatory	[55]
		ferulic acid	C_10_H_10_O_4_	anticarcinogenic, antioxidant, antimicrobial, hepatoprotective	[56]
		inulin	C_228_H_382_O_191_	kidney diseases, antibacterial	[57]
		isterine	C_21_H_30_O_9_	anti-inflammatory	[58]
		lupeol	C_30_H_50_O	anti-inflammatory, antitumor, antidiabetic, heart diseases	[59]
		hydroxybenzoic acid	C_7_H_6_O_3_	antioxidant	[60]
		monocaffeoyltartaric acid	C_13_H_12_O_9_	antioxidant, anti-inflammatory	[53]
		p-coumaric acid	C_9_H_8_O_3_	anti-inflammatory, antitumor, antioxidant	[54]
		protocatechuic acid	C_7_H_6_O_4_	anti-inflammatory	[55]
		stigmasterol	C_29_H_48_O	antioxidant, antitumor, antimicrobial, hepatoprotective	[56]
		syringin	C_9_H_10_O_5_	kidney diseases, anti-bacterial	[57]
		taraxalisin	67-kD glycoprotein	anti-inflammatory	[58]
		taraxasterol	C_30_H_50_O	anti-inflammatory, antitumor, antidiabetic, for heart diseases	[59]
		taraxerol	C_30_H_50_O	antioxidant	[60]
		taraxicinic acid	C_15_H_18_O_4_	anti-inflammatory	[58]
		tetrahydroridentin B	C_15_H_24_O	antimicrobial, anti-inflammatory	[55]
		umbelliferone	C_9_H_6_O_3_	anti-inflammatory, antihyperglycemic, antitumor, antibacterial, antifungal	[60]
		vanillic acid	C_8_H_8_O_4_	anti-inflammatory, antioxidant, cytoprotective, neurological disorders	[26,61]
		3-formyl indole	C_9_H_7_NO	anti-inflammatory, antitumor, antioxidant, antidiabetic	[29,62]
	*T. fomosanum*	beta-carotene	C_40_H_56_	antitumor, antioxidant, immunostimulant	[63]
		caffeic acid	C_9_H_8_O_4_	anti-inflammatory, antitumor, antioxidant	[64]
		caffeoylquinic acid	C_16_H_18_O_9_	anti-influenza	[64]
		caftaric acid	C_13_H_12_O_9_	antioxidant, antidiabetic	[64]
		chicoric acid	C_22_H_18_O_12_	antioxidant, anti-inflammatory, antidiabetic	
Root	*T. fomosanum*	chlorogenic acid	C_16_H_18_O_9_	anti-inflammatory	[64]
		luteolin-7-O-glucoside	C_21_H_20_O_11_	anti-inflammatory, anticancer, antidiabetic	[64]
		nicotinamide	C_6_H_6_N_2_O	anticancer, skin diseases	[29]
		protocatechuic acid	C_7_H_6_O_4_	anti-inflammatory	[55]
		stigmasterol	C_29_H_48_O	antitumor, antioxidant, antimicrobial	[29]
		syringic acid	C_9_H_10_O_5_	antimicrobial, antidiabetic, antitumor	[29,65]
		taraxafolide, taraxafolin-B	Substance SID: 275594471	antibacterial, anti-inflammatory	[29,66]
		vanillic acid	C_8_H_8_O_4_	antioxidant, cytoprotective	[29,67]
	*T. campylodes*	apigenin	C_15_H_10_O_5_	antioxidant, antitumor, antiviral, antibacterial, nervous, kidney diseases	[68,69]
		chicoric acid	C_22_H_18_O_12_	antidiabetic, antioxidant, anti-inflammatory	[70]
		chlorogenic acid	C_16_H_18_O_9_	anti-inflammatory	[70]
		isoquercitrin	C_21_H_20_O_12_	antitumor, antioxidant, antidiabetic, cardiovascular disorders	[68]
		luteolin	C_15_H_10_O_6_	antiviral, antidiabetic, anti-asthmatic, antitumor	[68]
		taraxacin	C_15_H_14_O_3_	liver and kidney disorders, antitumor	[71]
	*T. mongolicum*	ainsloside	C_37_H_62_O_16_	antioxidant, antitumor	[72]
		apigenin	C_15_H_10_O_5_	antitumor, antioxidant, anti-inflammatory	[72]
		baicalein	C_15_H_10_O_5_	anti-inflammatory, cardiovascular, respiratory and gastrointestinal disorders	[73]
		caffeic acid	C_9_H_8_O_4_	anti-inflammatory, antitumor, antioxidant	[72]
		chicoric acid	C_22_H_18_O_12_	antidiabetic, antioxidant	[72]
		chlorogenic acid	C_16_H_18_O	anti-inflammatory	[72]
		galacturonic acid	C_6_H_10_O_9_	anti-inflammatory, gastrointestinal disorders	[74]
		glucose	C_6_H_12_O_6_	ubiquitous energy source	[74]
		heperetin-5′-O-β-rhamno-glucoside	C_22_H_24_O_11_	antioxidant, antitumor	[73]
		kaempferol-3-glucoside	C_21_H_20_O_11_	anti-inflammatory, antioxidant, antiviral, antiallergic	[73]
		lutein	C_40_H_56_O_2_	anticarcinogenic, photo-protector, antioxidant, anti-inflammatory	[72]
		p-coumaric acid	C_9_H_8_O_3_	anticancer, antiulcer, antioxidant, anti-inflammatory, anti-mutagenic	[72]
		quercetin	C_15_H_10_O_7_	anti-inflammatory, antiallergic, antitumoral, antioxidant	[73]
	*T. coreanum*	caffeic acid	C_9_H_8_O_4_	anti-inflammatory, antitumor, antioxidant	[75]
		decursinol	C_14_H_14_O_4_	anti-inflammatory, analgesic, antineoplastic agent	[30]
		inositol	C_6_H_12_O_6_	nervous and metabolic disorders, antidiabetic	[30]
Root	*T. coreanum*	isoferulic acid	C_10_H_10_O_4_	antidiabetic	[30]
		pinoresinol	C_20_H_22_O_6_	hypoglycaemic agent, increase apoptosis	[30]
		syringaldehyde	C_9_H_10_O_4_	antioxidant	[30]
		taraxathin	C_40_H_56_O_3_	membrane stabilizer, flavoring agent	[75]
		taraxinositols A (1), B (2)	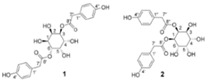 Not found in the literature		[30,76]
		taraxinol	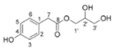 Not found in the literature		[30]
		vanillic acid	C_8_H_8_O_4_	antioxidant, cytoprotector	[30]
Leaves	*Taraxacum officinale*	anthraquinones	C_14_H_8_O_2_	antiviral, immunostimulatory, diuretic, laxative, antifungal	[77]
		apigenin-7-glucoside	C_21_H_20_O_10_	anticancer, antifungal	[78]
		β-branched glucomannan	C_24_H_42_O_21_	antidiabetic, laxative, hypocholesterolaemiant	[79]
		beta-sitosterol	C_29_H_50_O	reduction of benign prostatic hyperplasia and blood cholesterol levels	[54,80,81]
		betulin	C_30_H_50_O_2_	anticancer, antiobesity, antidiabetic	[82]
		caffeoyl glucoside	C_15_H_18_O_9_	antiviral, antitumor, antidiabetic, antifungal, antioxidant	[83]
		caffeoylmalic acid	C_13_H_12_O_8_	antidiabetic, antioxidant, antiproliferative, apoptotic effect	[84,85]
		chicoric acid	C_22_H_18_O_12_	antidiabetic, antioxidant	[51]
		cichorin	C_18_H_20_O_3_	antiparasitic activity	[55,86]
		dodecane	C_15_H_24_O	antimicrobial	[77]
		stafiatin	C_15_H_18_O_3_	anti-inflammatory, antitumor	[82]
		ferulic acid	C_10_H_10_O_4_	antioxidant, anti-inflammatory, neuroprotective, skin disease	[87]
		hesperidin	C_28_H_34_O_15_	anti-inflammatory, antitumoral, antihypertensive, antihyperlipidemic	[88]
		hydroxycinnamic acid	C_9_H_8_O_3_	antioxidant, skin protector	[51]
		kaempferol	C_15_H_10_O_6_	anti-tumoral, anti-inflammatory, antioxidant, respiratory diseases	[89]
		lettucenin A	C_15_H_12_O_3_	antifungal, antimicrobial	[90,91]
		lupeol acetate	C_32_H_52_O	anticancer, antidiabetic, anti-inflammatory and antiprotozoal	[54]
		lutein	C_40_H_56_O_2_	antitumoral, antioxidant, anti-inflammatory, photoprotector	[32,92]
		luteolin diglycoside	C_27_H_30_O_16_	antidiabetic	[83]
Leaves	*T. officinale*	monocaffeoyltartaric acid	C_13_H_12_O_9_	antioxidant, anti-inflammatory	[38]
		phytol	C_20_H_40_O	antioxidant, antimicrobial activity	[54]
		protocatechuic acid	C_7_H_6_O_4_	anti-inflammatory, antihyperglycemic, antioxidant, neuroprotector	[93]
		quercetine glycosides	C_21_H_19_O_12_	antioxidant, anti-inflammatory, cardiovascular disorders	[83,94]
		scopoletin	C_10_H_8_O_4_	antifungal, antitumoral	[95]
		sesquiterpene lactones	C_22_H_30_O_7_	anticancer, anti-inflammatory, antitumoral, antiviral, antibacterial, antifungal	[38]
		sinapic acid	C_11_H_12_O_5_	antioxidant, antimicrobial, anticancer, anti-inflammatory and antianxiety activity	[44]
		stigmasterol	C_29_H_48_O	antitumor, antioxidant, antimicrobial	[96,97]
		violaxanthin	C_40_H_56_O_4_	anti-inflammatory, antitumoral, antioxidant	[39]
	*T.* *coreanum*	chrysoeriol	C_16_H_12_O_6_	anti-inflammatory, antioxidant, antitumor effects	[98,99,100,101]
		daucosterol	C_35_H_60_O	antitumor, anti-inflammatory, antioxidant	[99,102,103]
		diosmetin	C_16_H_12_O_6_	antioxidant, antineoplastic, antitumoral, antimicrobial, antidepressive	[99]
		esculetin	C_9_H_6_O_4_	antioxidant, antidiabetic, antitumor	[99,104]
		luteolin	C_15_H_10_O_6_	antiviral, antidiabetic, antiasthmatic, antitumoral	[99,105]
		luteolin-7-O-glucoside	C_21_H_20_O_11_	anti-inflammatory, anticancer, antidiabetic	[99,105]
		sitosterol	C_29_H_50_O_10_	anti-inflammatory, antioxidant, antidiabetic, antianxiety, liver-protector	[99]
		taraxasteryl acetate	C_32_H_52_O_2_	anticancer, anti-inflammatory, antidiabetic, anticaught and lung diseases	[99,106]
		vanilinic acid	C_8_H_8_O_4_	antioxidant, cytoprotector	[30]
	*T. campylodes*	chlorogenic acid	C_16_H_18_O_9_	anti-inflammatory	[70]
		chicoric acid	C_22_H_18_O_12_	antidiabetic, antioxidant	[70]
	*T. mongolicum*	caffeic acid	C_9_H_8_O_4_	anti-inflammatory, antitumor, antioxidant	[107]
		esculetin	C_9_H_6_O_4_	antioxidant, antidiabetic, antitumor	[108]
		isoetin	C_15_H_10_O_7_	antioxidant, anti-inflammatory	[109]
		inositol	C_6_H_12_O_6_	nervous and metabolic disorders, antidiabetic	[21]
		luteolin 7-*O*-β-D-glucopyranoside	C_21_H_20_O_11_	anti-inflammatory, anticancer, antidiabetic	[107]
		stigmasterol	C_29_H_48_O	anti-inflammatory	[110]
		quercetin	C_15_H_10_O_7_	anti-inflammatory, antiallergic, antitumor, antioxidant	[111]
		taraxasterol	C_30_H_50_O	anti-inflammatory, antitumor, antioxidant, lung disease	[110]
Flowers	*T. officinale*	arnidiol	C_30_H_50_O_2_	anti-inflammatory	[55,94]
		beta-carotene	C_40_H_56_	antitumor, antioxidant, immunostimulant	[32]
		chlorogenic acid	C_16_H_18_O	anti-inflammatory	[44,51,94]
		chrysoeriol	C_16_H_12_O_6_	antineoplastic agent, antioxidant, antimicrobial	[83]
		heneicosane	CH_3_(CH_2_)_19_CH_3_	antimicrobial, antioxidant, analgesic, antipyretic	[60]
		luteoline 7-O-glucoside	C_21_H_20_O_11_	anticancer, antidiabetic antioxidant	[112,113]
		hydroxycinnamic acid	C_9_H_8_O_3_	antioxidant, skin protector	[53,94]
		monocaffeoyltartaric acid	C_13_H_12_O_9_	antioxidant, anti-inflammatory	[114]
		pectin	C_6_H_10_O_7_	anti-inflammatory, antibacterial, antioxidant, antitumor activities	[115]
		routine	C_27_H_30_O_16_	antimicrobial	[83]
		quercetin	C_15_H_10_O_7_	anti-inflammatory, antihypertensive, anti-obesity	[83]
		stigmasterol	C_29_H_48_O	anti-inflammatory	[96]
		tricosane	C_23_H_48_	antimicrobial	[37]
	*T.* *formosanum*	not found in literature			
	*T. mongolicum*	caffeic acid caffeic acid	C_9_H_8_O_4_	anti-inflammatory, antitumor, antioxidant	[52]
		caffeoylquinic acid	C_16_H_18_O_9_	anti-influenza	[52]
		caftaric acid	C_13_H_12_O_9_	antioxidant, antidiabetic	[52,116]
		chicoric acid	C_22_H_18_O_12_	antioxidant, anti-inflammatory, antidiabetic	[52]
		chlorogenic acid	C_16_H_18_O_9_	anti-inflammatory	[52,117]
		delphinidin 3-*O*-glucoside	C_21_H_21_O_12_	antitumor, hypolipidemic, endothelial protective	[116]
		luteolin	C_15_H_10_O_6_	antiviral, antidiabetic, antiasthmatic, antitumor	[52,117]
	*T. coreanum*	adenosine	C_10_H_13_N_5_O_4_	neuromodulator, reduce tissue injury and promote repair	[118]
		astragalin	C_21_H_20_O_11_	anti-inflammatory, antitumor, antioxidant, neuroprotective	[118]
		chicoric acid	C_22_H_18_O_12_	antioxidant, anti-inflammatory, antidiabetic	[75]
		isoquercitine	C_21_H_20_O_12_	antiviral, anti-inflammatory, antioxidant	[118]
		luteolin	C_15_H_10_O_6_	antiviral, antidiabetic, anti-asthmatic, antitumor	[75]
Entire vegetal product	*T. officinale*	alfa and beta-amyrin	C_30_H_50_O	anti-inflammatory, analgesic, gastroprotective, hepatoprotective, antihyperglycemic, anti-obesity effects	[58,59]
		chicoric acid	C_22_H_18_O_12_	anti-inflammatory, antioxidant	[83]
		monocaffeoyltartaric acid	C_13_H_12_O_9_	antioxidant, anti-inflammatory	[114]
		chlorogenic acid	C_16_H_18_O_9_	anti-inflammatory, antioxidant, antiviral, antibacterial, antihypertensive	[83]
		ferulic acid	C_10_H_10_O_4_	anticarcinogenic, antioxidant, antimicrobial, hepatoprotective	[51]
		taraxacin	C_15_H_14_O_3_	anti-inflammatory and anticancer	[16]
		vanillic acid	C_8_H_8_O_4_	antioxidant, cytoprotector	[30]
	*T. coreanum*	adenosine	C_10_H_13_N_5_O_4_	neuromodulator, reduce tissue injury and promote repair	[118]
		guanosine	C_10_H_13_N_5_O5	neuroprotective, reduce neuroinflammation and oxidative stress	[118]
		luteolin	C_15_H_10_O_6_	antiviral, antidiabetic, anti-asthmatic, antitumor	[75]
		5′-methyl-thioadenosine	C_11_H_15_N_5_O_3_S	anti-inflammatory, liver damage	[118]
		quercetin	C_15_H_10_O_7_	anti-inflammatory, antihypertensive, anti-obesity	[75]
		taraxinic acid	C_15_H_18_O_4_	anti-inflammatory	[75]
	*T. campylodes*	austricin 8-O-β-D-glucopyranoside	C_21_H_28_O_9_	anti-inflammatory, anticancer	[119,120]
		caftaric acid	C_13_H_12_O_9_	antioxidant, antidiabetic	[119]
		3,5-di-O-caffeoyl-quinic acid	C_25_H_24_O_12_	antimutagenic	[119]
		chichoric acid	C_22_H_18_O_12_	antidiabetic, antioxidant	[119]
		chlorogenic acid	C_16_H_18_O_9_	antioxidant, anti-inflammatory, antiviral, antibacterial, antihypertensive	[119]
		inositol	C_30_H_30_O_12_	nervous and metabolic disorders, antidiabetic	[119]
		luteoline 7-O-glucoside	C_21_H_20_O_11_	anticancer, antidiabetic antioxidant	[119]
		taraxinic acid	C_15_H_18_O_4_	anti-inflammatory	[119]
	*T. mongolicum*	caffeic acid	C_9_H_8_O_4_	anti-inflammatory, antitumor, antioxidant	[52]
		caffeoylquinic acid	C_16_H_18_O_9_	anti-influenza	[52]
		caftaric acid	C_13_H_12_O_9_	antioxidant, antidiabetic	[52,116]
		chicoric acid	C_22_H_18_O_12_	antioxidant, anti-inflammatory, antidiabetic	[52]
		chlorogenic acid	C_16_H_18_O_9_	anti-inflammatory	[52,117]

Biel et al. [121] analyzed the mineral composition of 224 samples of dandelion leaves and concluded that it is rich in Ca (0.67 g/100 g^−1^ DM), Fe (14.10 g/100 g^−1^ DM), Mg (0.24 g/100 g^−1^ DM) and, particularly, in K (6.51 g/100 g^−1^ DM). Biotin, calcium, choline, gluten, inositol, inulin, iron, lactucopicrin, linolenic acid, magnesium, niacin, phosphorus, potash, proteins, resin, sulfur, vitamins A, B_1_, B_2_, B_5_, B_6_, B_9_, B_12_, C, E, and P, and zinc are also reported [122].

## 3. Biomedical Effects

The first mentioned use of *Taraxacum officinale* as a medicine is in the works of the Arabian physicians of the 10th and 11th centuries, who speak of it as a sort of wild Endive under the name of “Taraxcacon”. In the past, dandelion roots and leaves were used to treat liver problems [123]. Throughout history, various cultures have harnessed the healing potential of dandelion. Native American tribes used it to treat a range of ailments, including kidney disease, skin conditions, and digestive issues [124]. In Europe, dandelion has been valued for its diuretic properties, particularly in France [123].

Dandelion has a long history of use in traditional Indian medicine, particularly in the Himalayan regions. In Kashmir, for example, a paste made from boiled leaves, salt, and turmeric is applied to bone fractures [125]. Elsewhere in India, the plant is consumed as a leafy green vegetable, and its roots are used to treat liver and kidney ailments, as well as snakebites [126].

Modern medicine has also begun to recognize the therapeutic potential of dandelion. It has been used to improve liver function, lower cholesterol, reduce blood pressure, aid in weight loss, and alleviate gallbladder and kidney disorders [127,128,129].

Traditional Chinese medicine considers dandelion a non-toxic herb with valuable properties, including choleretic, anti-inflammatory, diuretic, and anti-rheumatic effects. Studies have shown that dandelion root and leaf extracts can lower lipid levels, reduce oxidative stress, and protect against liver damage [130,131,132]. Additionally, taraxasterol, a compound isolated from the Chinese medicinal herb *Taraxacum*, has demonstrated anti-arthritic effects in animal models [133].

Modern medicinal uses of dandelion are focused on digestive health, taking into account that it improves digestion and relieves digestive disorders, aiding in the breakdown of food and cholesterol reduction, supports liver and gallbladder function, aids in detoxification and supports and relieves gallbladder-related issues. Beyond its digestive benefits, dandelion also exhibits diuretic properties. By increasing urine production, it helps eliminate excess fluids and may alleviate conditions like kidney stones and other urinary tract problems. Some studies suggest that dandelion may even contribute to maintaining healthy blood pressure.

The therapeutic effects of dandelion are attributed to its diverse range of bioactive compounds, including polyphenols, powerful antioxidants with anti-inflammatory properties, sesquiterpene lactones: compounds with anti-inflammatory, anti-tumor, and antimicrobial properties, inulin: a soluble fiber that promotes gut health, vitamins, and minerals: essential nutrients that support overall health.

By understanding the traditional and modern uses of dandelion, we can appreciate its potential as a valuable medicinal plant. However, it is important to consult with a healthcare specialist before using dandelion for therapeutic purposes, especially if you are pregnant, breastfeeding, or taking medications.

Primarily valued as a medicinal plant, dandelion has been employed for various therapeutic applications. The European Pharmacopoeia (2005) and the Committee for Herbal Medicinal Products of the European Medicines Agency have authorized the use of the entire dandelion plant, including its roots, in therapeutic treatments [134]. Dandelion is considered non-toxic and possesses considerable biological activity [135].

The leaves of *T. officinale*, often combined with other plants, have been used to treat liver, kidney, skin, and even cancer [136]. Dandelion infusion has shown potential benefits in managing kidney stones, likely due to disinfectant properties and the presence of saponins [137]. With low levels of essential oils and tannins, *T. officinale* has a high feed value [121]. Plant-derived polyphenols have demonstrated anti-inflammatory properties both in laboratory and live animal studies, suggesting their potential as therapeutic agents for various acute and chronic conditions [138,139,140]. These compounds exert their anti-inflammatory effects by reducing oxidative stress and modulating the expression of pro-inflammatory genes, such as cytokines, lipoxygenase, nitric oxide synthases (iNOS), and cyclooxygenase (COX-2) [141,142]. Polyphenols also offer cardiovascular protection by lowering blood pressure, reducing oxidative stress, and inhibiting LDL oxidation, which contributes to endothelial protection [143]. Dandelion leaf extract has been shown to reduce serum lipids and insulin resistance in a high-fat diet-induced model of fatty liver disease [78].

In a human study, hydroethanolic extracts of fresh *T. officinale* leaves significantly increased urinary frequency and volume [144]. Bile salts and acids are strongly bound by vegetable fibrous tissue [145,146], and consumption of dietary fiber can promote human health through its specific physicochemical functions. Gomez et al., in 2018, showed that chenodeoxycholate was bound maximum as compared with other bile acids salts (glycochenodeoxycholate, glycodeoxycholate, deoxycholate, cholate, and glycocholate) to dandelion whole-leaf samples grown in Texas and New Jersey [39].

Dandelion polysaccharides have exhibited strong immune-regulating, anti-inflammatory, and antioxidant properties. *Taraxacum* extracts have demonstrated the ability to lower triglyceride, LDL cholesterol levels, and total cholesterol while simultaneously raising HDL cholesterol levels. Additionally, its polysaccharides have been shown to reduce the size of atherosclerotic lesions and necrotic cores in the aortic sinus and increase collagen content [147]. Dandelion flower water syrup has been found to decrease vasoconstricting prostanoids in the thoracic arteries of obese rats [148]. Moreover, dandelion polysaccharides have been shown to mitigate liver damage caused by CCl_4_ by regulating inflammatory responses and reducing oxidative stress in rats [149].

Previous research has indicated that dandelion extracts can stimulate the production of multiple anti-inflammatory cytokines [150]. More recently, Hao et al. (2024) [151] reported that both crude extracts and specific compounds derived from *Taraxacum* plants exhibit a range of pharmacological properties, including antimicrobial, antitumor, anti-inflammatory, antioxidant, hepatoprotective, and blood glucose and lipid-regulating effects.

### 3.1. Anti-Inflammatory Activity

A Complex Biological Response—inflammation is a natural defense mechanism triggered by harmful stimuli, such as infections, injuries, or toxins. This response aims to restore the body’s balance by initiating a series of physiological changes [152]. The classic signs of inflammation include redness, swelling, heat, pain, and loss of function [153]. These symptoms arise from a complex interplay of cellular and molecular events involving various immune cells and signaling molecules.

Immune cells, including neutrophils and macrophages, are key players in the body’s inflammatory response. When these cells are activated, they release a range of inflammatory substances, such as cytokines (e.g., tumor necrosis factor = TNF-α, interleukin-1 beta = IL-1β, and interleukina-6 = IL-6), chemokines, prostaglandins, leukotrienes, and reactive oxygen species (ROS) [154,155]. These mediators induce changes in blood vessels, such as vasodilation and increased permeability, allowing immune cells to migrate to the site of injury or infection.

Histamine, a potent inflammatory mediator released by mast cells, contributes to various allergic responses, including asthma. It triggers bronchoconstriction, vasodilation, and increased mucus production in the airways [156].

Dandelion, *T. officinale*, exhibits significant anti-inflammatory properties, making it a promising therapeutic agent for various inflammatory conditions. The plant’s anti-inflammatory effects can be attributed to several mechanisms, including the inhibition of pro-inflammatory cytokines. Dandelion extracts have been shown to reduce the production of pro-inflammatory cytokines such as TNF-α and IL-1β, which play a crucial role in inflammatory processes; modulation of oxidative stress: dandelion possesses potent antioxidant properties, which help to neutralize harmful free radicals and reduce oxidative stress-induced inflammation; suppression of immune cell activation: dandelion extracts can inhibit the activation and migration of immune cells, such as neutrophils and macrophages, to the site of inflammation; inhibition of lipid mediators: the plant’s compounds can prevent the production of inflammatory lipid mediators, including prostaglandins and leukotrienes.

Scientific evidence suggests that dandelion leaf extract can inhibit the production of TNF-α and IL-1β in rat astrocytes, highlighting its potential to alleviate neuroinflammation [157]; other results suggested that aqueous extracts of dandelion leaves may inhibit TNF-a production by suppressing IL-I production and the plant exhibits anti-inflammatory activity in the central nervous system [158]. Furthermore, dandelion has been shown to reduce the severity of pancreatitis by inhibiting the production of pro-inflammatory cytokines and promoting the expression of heat shock proteins [159], while dandelion root extracts have been found to inhibit the production of leukotriene B4, a potent inflammatory mediator, by leukocytes [57].

Dandelion flower extracts, particularly rich in luteolin and luteolin-7-O-glucoside, exhibit anti-inflammatory properties. In laboratory studies, these extracts suppressed the production of nitric oxide and prostaglandin E2, key inflammatory molecules, in immune cells. This suggests dandelion’s potential to modulate inflammatory responses without introducing cytotoxicity [27]. Additionally, animal studies suggest that dandelion extracts may help protect against induced inflammation. They have been shown to reduce the severity of pancreatitis, an inflammatory condition of the pancreas, while also decreasing inflammatory markers and increasing stress-protective proteins [159].

Hudec et al. [160] investigated the antioxidant and antiradical activities of dandelion, along with the total content of various phenolic compounds. The authors analyzed both the roots and aerial parts of the plant after treating them with a polyamine inhibitor and a phenol biosynthesis stimulator. The results showed that both treatments increased the plant’s antioxidant activity.

Jeon et al. extracted dandelion with ethanol and found that the extract had several beneficial effects. It reduced intracellular reactive oxygen species, inhibited angiogenesis, and reduced inflammation in animal models. The extract also inhibited the production ofnitric oxide and the expression of inflammatory enzymes like iNOS and COX-2 in immune cells [161].

Liu et al. [162] investigated the effects of taraxasterol, a compound found in dandelion, on inflammation and cholesterol metabolism. They discovered that taraxasterol inhibited the production of inflammatory cytokines and the activation of a key inflammatory signaling pathway. It also disrupted the formation of lipid rafts, which are involved in inflammatory signaling. Additionally, taraxasterol was found to activate a pathway that promotes cholesterol efflux from cells.

A 2010 study demonstrated that dandelions could protect against acute lung injury induced by lipopolysaccharide (LPS) in mice. Dandelion was shown to reduce the production of inflammatory cytokines, such as TNF-α and IL-6, in lung fluid and decrease the number of inflammatory cells and lung weight [163]. Furthermore, dandelion increased the activity of antioxidant enzymes like superoxide dismutase and reduced the activity of myeloperoxidase, an enzyme involved in inflammation. These effects were attributed to the compound luteolin [164]. Further research has indicated that dandelion leaves can reduce the expression of inflammatory enzymes iNOS and COX-2 and the production of inflammatory mediators in a dose-dependent manner. This effect is achieved by inhibiting the MAPK signaling pathway [150].

Aqueous and methanolic extracts of *Taraxacum* leaves have been shown to alleviate oxidative stress and the inflammatory response in LPS-stimulated RAW 264.7 macrophages [165]. The combined use of both extracts decreased the levels of NO and MDA (malondialdehyde) by downregulating the expression and production of iNOS. This effect was achieved through the inhibition of the phosphatidylinositol-3-kinase (PI3K)/Akt pathways and the disruption of nuclear factor kappa B (NF-kB) translocation in LPS-stimulated cells. Additionally, there was an increase in the levels of antioxidant enzymes, including catalase, superoxide dismutase, glutathione peroxidase, and glutathione reductase. The methanolic extract of *Taraxacum* demonstrated significantly higher antioxidative activity compared to the aqueous extract, which could be related to the higher phenolic content, including luteolin and chicoric acid.

In 2010, Awortwe et al. [166] (see Table 3) explored the anticholinergic and anti-inflammatory properties of *T. officinale* leaf extract (TOLE) in guinea pigs sensitized with ovalbumin (OA). Their study revealed that TOLE significantly inhibited tracheal contraction in response to both acetylcholine and OA, indicating an anticholinergic effect. Furthermore, TOLE led to a decrease in the number of monocytes, lymphocytes, and neutrophils in the blood of guinea pigs, suggesting anti-inflammatory activity and potential benefits in alleviating allergic responses.

A 2013 study by Awortwe displayed the anti-inflammatory effects of *T. officinale* ethanolic leaf extract on pulmonary vascular permeability and histamine receptors in guinea pigs sensitized with ovalbumin. The extract significantly reduced histamine-induced contractions in isolated guinea-pig ileum, indicating antihistamine activity. Furthermore, the extract alleviated lung inflammation, characterized by reduced perivascular edema, smooth muscle hypertrophy, and infiltration of inflammatory cells like eosinophils and basophils [167].

Alongside luteolin and chicoric acid, taraxasterol is also involved in anti-inflammatory activity. It led to the downregulation of the iNOS and COX-2 expression and production, mediated through the inhibition of ERK1/2 and phosphorylation [168]. Taraxasterol, isolated from *T. officinale*, exhibits protective effects in a murine model of endotoxic shock induced by lipopolysaccharides. It achieves this by modulating the secretion of inflammatory cytokines and significantly reducing serum levels of TNF-α, IFN-γ, IL-1β, IL-6, NO, and PGE2 in mice [169]. Anti-inflammatory effects of polysaccharides isolated from *T. officinale* LPS-stimulated murine macrophages RAW 264.7 were also reported. Taraxasterol, a compound found in *Taraxacum* herba, has been shown to inhibit inflammation by regulating the NF-κB signaling pathway and activating the Nrf2 pathway, which promotes antioxidant activity [170].

Clinical studies have demonstrated the effectiveness of *Taraxacum* herba in treating acute mastitis. A study by Jun Qi et al. found that 57 out of 58 patients treated with *Taraxacum* herba experienced relief from symptoms such as redness, swelling, fever, and pain, with a significant reduction in body temperature. These results highlight the potential therapeutic benefits of *Taraxacum* herba in treating inflammatory conditions [176].

Dandelion extracts have shown promising anti-inflammatory and antioxidant properties. They can protect cells from oxidative stress by activating the Nrf2 pathway, which upregulates the expression of the antioxidant enzyme HO-1. Additionally, dandelion extracts can inhibit the NF-κB and Akt signaling pathways, which are involved in inflammation. This leads to a reduction in the production of inflammatory mediators like nitric oxide, iNOS, and TNF-α. In animal models, dandelion extracts have demonstrated protective effects against inflammation-induced diseases like arthritis by lowering the levels of inflammatory cytokines and shielding cells from oxidative damage. Additionally, dandelion polysaccharides have been shown to inhibit the expression of inflammatory factors such as COX-2, TNF-α, IL-6, and IL-1β in LPS-stimulated cells [177].

The treatment with this isolated compound significantly alleviated paw swelling and arthritis index. Improvements were observed in synovial hyperplasia, cartilage, and bone destruction, and the inflammatory cell infiltration into the joint-space narrowing was observed. Taraxinic acid β-D-glucopyranosyl ester, a sesquiterpene lactone isolated from *T. officinale* leaves, is a relatively potent inducer of Nrf2 transactivation [38]. Additionally, the spleen and thymus indexes were strongly reduced, suggesting that taraxasterol may help in the recovery of the hyperfunctioning immune organs without causing damage and the serum levels of some pivotal inflammatory mediators that contribute to the clinical manifestations of rheumatoid arthritis, namely TNF-a, IL-1b and PGE2, were significantly decreased. Therefore, taraxasterol was able to inhibit bone destruction in an animal model of rheumatoid arthritis. Also, taraxasterol has been shown to have anti-apoptotic effects by activating the Bax/Bcl-2 signaling pathway. This pathway is often dysregulated in autoimmune disorders, suggesting a potential therapeutic role for taraxasterol in these conditions [171].

In recent clinical studies, *Taraxacum* herba has shown promise in treating acute mastitis. When combined with minimally invasive catheter drainage, it has been found to be less traumatic, lead to quicker recovery, and have no impact on breastfeeding [178]. Additionally, when combined with cefradine, *Taraxacum* herba has been shown to significantly reduce levels of inflammatory markers such as IL-6, PCT, ICAM-1, and TNF-α [179].

A *Taraxacum* extract obtained from aerial parts demonstrated antioxidant activity in vivo [172]. The treatment of turpentine-induced acute inflammation in rats with this extract strongly reduced oxidative stress by decreasing MDA, thiols, and nitrites/nitrates levels. Moreover, the administration of the *Taraxacum* extracts also decreased the expression of the NF-jB signaling pathway. Recent research by Hee et al. [173] has demonstrated that ethanol extracts of *Taraxacum* herba effectively inhibited the production of nitric oxide and inflammatory factors such as COX-2, TNF-α, IL-6, and IL-1β. This suggests that dandelion may have potential therapeutic benefits in alleviating intestinal inflammation.

In another study by Bingjie et al. [174], dandelion extract was found to protect against liver injury induced by acetaminophen in a mouse model. The extract was shown to alleviate cellular damage, activate the Nrf2/HO-1 pathway, inhibit the JNK signaling pathway, and reduce apoptosis. These findings further support the anti-inflammatory and liver-protective properties of dandelion.

LPScan cause inflammation by impairing the brain endothelial barrier function and increasing the blood–brain barrier (BBB) permeability. In their study, Han et al. (2024) treated bEnd.3 cerebral vascular endothelial cells with *Taraxacum coreanum NAKAI* extract followed by LPS exposure and the effects on trans endothelial electrical resistance (TEER) values, pro-inflammatory cytokine production, and expression of proteins related to inflammatory responses and tight junction integrity were assessed. The treated group suppressed the protein’s expression related to the nuclear factor-κB pathway. Taken together, *T. coreanum* attenuates LPS-induced neuroinflammatory responses by regulating NF-κB activation, which may contribute to protecting against BBB disruption [180].

All these findings highlight the potential of dandelion as a natural anti-inflammatory agent. Further research is needed to fully elucidate the underlying mechanisms and to explore its therapeutic applications in various inflammatory disorders. The European Commission and the British Herbal Pharmacopoeia recommended the following range of doses for dandelion: fresh leaves 4–10 g daily, dried leaves 4–10 g daily, 2–5 mL of leaf tincture, three times a day, fresh leaf juice, 1 teaspoon twice daily, fluid extract 1–2 teaspoon daily Fresh roots 2–8 g daily, dried powder extract 0.25–1 g four times a day [135]. *Note*: It is crucial to consult with a healthcare professional before using dandelion or any herbal supplement, especially if you have underlying health conditions or are taking medications.

### 3.2. Antiviral Activity

Crude extracts of dandelion have been screened for their in vitro antimicrobial properties. Extracts of *Taraxacum farinosum* and *Taraxacum mirabile* [181] demonstrated in vitro antiviral effects against the Herpes Simplex Virus Type 1 (HSV-1), and results of Sengul et al. in 2009 [182] show that dandelion has antimicrobial activity, which has been attributed to its flavones. This antimicrobial activity might be mediated by the antioxidant activity of the flavones, but his hypothesis requires confirmation. Polyphenols are well known for their antiviral potential [183,184,185,186]. Their presence in medicinal plants and some other herbs is extensive, and due to their biological and pharmacological activity, they are recommended as therapeutic sources with medicinal and pharmaceutical proficiency [187,188].

There are studies that highlight the potential antiviral properties of dandelion, particularly against flaviviruses like yellow fever virus, dengue virus, and hepatitis C virus (HCV).

Rodriguez-Ortega et al. [189] used dandelion leaves, particularly those collected at a later stage of growth, to test antiviral activity against flavivirus, using the 17D vaccine strain of yellow fever virus. Dandelion extracts have shown antiviral activity against the yellow fever virus. Compounds like caffeic acid, V-taraxasteryl acetate, taraxasteryl acetate, taraxerol, and V-taraxasterol are responsible for the observed antiviral activity. Rehman et al. [190] used methanol extract to demonstrate that dandelion leaves are able to inhibit the replication of HCV by targeting the NS5B polymerase protein. The obtained results revealed that common dandelion leaf extract potentially blocked viral replication and NS5B gene expression without toxic effects on normal fibroblast body cells.

Flores-Ocelotl et al.’s [191] research demonstrated that methanolic extracts exhibited antiviral proprieties. Compounds present in *T. officinale,* like luteolin and caffeoylquinic acid derivatives, may contribute to this activity. Dengue virus (DENV), a member of the Flaviviridae family, causes dengue fever and more severe manifestations like dengue hemorrhagic fever and dengue shock syndrome [192].

Tran et al. [193] revealed that infection of the lung cells using SARS-CoV-2 pseudotyped lentivirus was efficiently prevented by the extract of water-based dandelion. Effective inhibition of protein-protein interaction between the human virus cell entry receptor ACE2 and SARS-CoV-2 spike, including five relevant mutations, by water-based dandelion extracts, was shown in vitro using human kidney (HEK293) and lung (A549) cells, overexpressing the ACE2 and ACE2/TMPRSS2 protein, respectively. Another remedy composed of dried fruits from three plant species—*Emblica officinalis*, *Terminalia bellerica*, and *Terminalia chebula*—Triphala, was also tested on lung epithelial cells (A549) induced by CoV2-SP [194]. Nanoparticles loaded with *Triphala* extract, named “nanotriphala”, as a drug delivery system, determined significantly reduced cytokine release (IL-6, IL-1β, and IL-18) and suppressed the expression of inflammatory genes (IL-6, IL-1β, IL-18, and NLRP3). Mechanistically, nanotriphala and its active compounds decreased the expression of inflammasome machinery proteins (NLRP3, ASC, and Caspase-1).

Polyphenols, like chlorogenic acid and caffeic acid, were identified by the UPLC-PDA-ESI- MS/MS method in *Taraxacum campylodes* G.E. Haglund “dandelion”, a medicinal plant consumed in traditional medicine in Trujillo, Peru and was evaluated antiviral activity in silico against the viral proteins NS2B/NS3 (DENV-2), NS5B (HCV), and ICP27 (HSV-1) by molecular docking using Chimera 1.16 software and molecular interaction by Maestro 13.1 software to identify the position and type of interaction [195] (see Table 4). The antiviral activity of chlorogenic acid stood out against DENV-2 and HCV.

### 3.3. Antimicrobial Activity

Antimicrobial peptides (AMPs) are essential components of the innate immune system in many organisms [196]. Among these, plant-derived peptide protease inhibitors (PPIs) play a crucial role in plant defense. These PPIs can directly attack pathogens or modulate the plant’s defense response by inhibiting the growth of bacteria, fungi, and protozoa. They achieve this by interfering with essential biochemical processes or by altering the permeability of pathogen cell membranes (see Figure 6). Studies on various plant extracts, including those from the Asteraceae family, have demonstrated promising in vitro antimicrobial activity against a broad spectrum of microorganisms. This indicates that members of this family could serve as a valuable source of natural antimicrobial compounds.

Oseni and Yussif’s study [197] investigated the antibacterial properties of ethanol and aqueous extracts of dandelion against common bacteria: *Escherichia coli*, *Klebsiella pneumoniae*, *Pseudomonas aeruginosa*, and *Staphylococcus aureus*. They found that both extracts exhibited antibacterial activity, with the ethanol extract being more potent than the aqueous extract. The activity was concentration-dependent, with higher concentrations leading to greater inhibition. Among the tested bacteria, *E. coli* was most susceptible to the extracts. In their research, Tian et al. [198] have synthetised three colloids of silver nanoparticles (AgNPs) with different biological materials, such as *Arctium lappa* fruit, *Solanum melongena* leaves, and *T. mongolicum* leaves. The three botanical AgNPs had the strongest bacteriostatic effect against Xoo strain C2 (rice bacterial leaf blight caused by *Xanthomonas oryzae pv. oryzae*) at 20 μg/mL with the inhibition zone between 16.5 mm and 12.4 mm, while bacterial numbers in a liquid broth decreased. Goyal et al. [199] demonstrated that dandelion extracts, particularly those from diploid cytotypes, exhibited significant antibacterial activity against a range of bacteria, including *Bacillus subtilis*, *S. aureus*, *E. coli*, *K. pneumoniae*, and *P. aeruginosa*.

**Figure 6 ijms-26-00450-f006:**
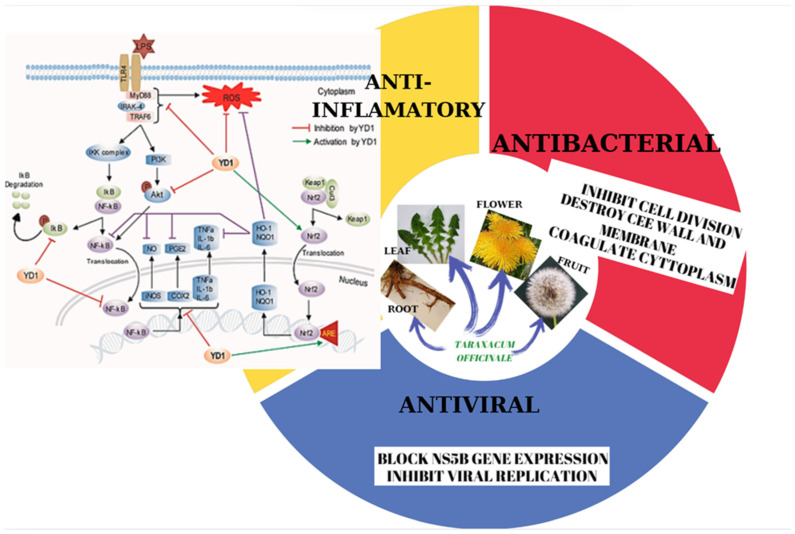
The described anti-inflammatory, antibacterial, and antiviral mechanisms of *Taraxacum* extracts (adapted after [200]).

Astafieva et al. [201] encountered three antimicrobial peptides designated ToAMP1, ToAMP2 and ToAMP3. These peptides were purified from *T. officinale* flowers, and their amino acid sequences were determined. The peptides were shown to exhibit strong antimicrobial activity against both fungal and bacterial pathogens, therefore making them promising candidates for biotechnological and medicinal applications.

Ionescu et al. [202] determined the antimicrobial activity of hydroalcoholic extracts of dandelion. The activity was tested using a serial dilution method on the following bacterial strains: *S. aureus*, *E. coli*, and *Salmonella abony*. Antimicrobial activity against *E. coli* and *S. abony* was observed, while no activity was found against *S. aureus.*

A novel nutraceutical product combining curcuminoids, vitamins, minerals, and phytochemicals from various plants, including dandelion (*T. officinale*), was evaluated in 2021 by Răducanu et al. [203]. This product demonstrated probiotic properties, promoting the growth of beneficial gut bacteria without harming the cells. It showed stronger antibacterial activity against Gram-positive bacteria like *S. aureus* compared to Gram-negative bacteria (*E. coli*) and fungi (*Candida albicans*). In another study, ZnO nanoparticles were synthesized using an extract of *T. officinale* (see Table 5) roots [204]. These nanoparticles displayed significant antibacterial activity against a series of bacteria, including *P. aeruginosa*, *S. aureus*, *E. coli*, *K. pneumoniae*, and *S. pneumoniae*. Additionally, the ZnO nanoparticles demonstrated antioxidant properties.

These findings suggest that dandelion-derived compounds and nanoparticles have potential applications in various fields, including medicine and food science. However, further research is needed to fully understand their mechanisms of action and to develop safe and effective products.

Hydroalcoholic extract obtained by refluxing and macerated fluids of dandelion leaves, radix, and flowers in ethyl alcohol was highly active against *S. aureus* and moderately active against *E. coli*. The smallest inhibitory concentration was revealed at radix and flower hydroalcoholic extract obtained by refluxing in ethanol:water 70:30 (*v:v*), tested on *P. aeruginosa* [205]. Enteshari Najafabadi et al. [206] demonstrated that extracted polysaccharides from *T. officinale* (2%) were significantly more effective in increasing the growth of *Lactobacillus rhamnosus* as an indigenous probiotic in comparison to two commercial prebiotics, i.e., inulin and dextrose. The maximum values of the antibacterial properties against four bacteria (*E. coli*, *P. aeruginosa*, *S. aureus*, and *B. subtilis*) were related to polysaccharide isolated from *T. officinale*, major constituents being cyanidin-3-O-β-glucoside, N-acetylcysteine, and glutamic acid in *T. officinale*, which may play a major role in the biological properties of this plant.

**Table 5 ijms-26-00450-t005:** Antimicrobial activity of *T. officinale* = *T.o*; *T. mongolicum* = *T.m*.

Vegetal Organ/Phytocompounds	Type of Experiment	Results	Reference
*T. o.* ethanolic and aqueous leaf extracts	*E. coli*, *K. pneumoniae*, *P. aeruginosa*, *S. aureus*	ethanolic leaf extract at concentrations of 200 mg/mL and 100 mg/mL inhibited only *E. coli* and *S. aureus*, while the 50 mg/mL ethanolic leaf extract inhibited only *E. coli*. The aqueous extract showed inhibition against *E. coli* at concentrations of both 200 mg/mL and 100 mg/mL	[197]
*T. m.* leaves water extract	silver nanoparticles (AgNPs) were synthesized with biological material tested on *Xanthomonas oryzae pv. oryzae* (Xoo)	strong bacteriostatic against the Xoo strain at 20 µg/mL with an inhibition zone of 12.4 mm, while bacterial numbers in a liquid broth (measured by OD600) decreased by 65.60%	[198]
*T. o.* ethanol extract of leaves from 3 cytotypes	*B. subtilis*, *S. aureus*, *E. coli*, *K. pneumonia*, *P. aeruginosa*	diploid cytotype, with the best cichoric acid concentration, exhibited the highest antibacterial activities	[199]
*T. o.* flowers and seeds polypeptides	pathogenic fungi: *A. niger*, *A. fumigatus*, *P. chrysogenum*, *F. oxysporum*, *C. albicans*	three mg/mL of dandelion flower concentrate increased the survival rate of the yeast test culture CFU by more than 1.5 times, while the dandelion seed extract was inactive. None of the peptides had activity against *A. niger*, ToAMP3 had moderate antifungal action against *P. chryzogenum* and a weak against *A. fumigatus*	[201]
*T. o.* Ethanol extract of leaves	bacterial strains—*S. aureus*, *E. coli*, *S. abony enterica*	exhibited antimicrobial activity against *E. coli* and *S. abony enterica*, but no antimicrobial activity against *S. aureus*	[202]
*T. o.* root extract	*P. aeruginosa*, *S. aureus*, *E. coli*, *K. pneumoniae*, *S. pneumoniae*	efficient capping or reducing agent for the synthesis of nanoparticles, which can be developed as an antibacterial agent that is highly specific for a broad range of microorganisms in order to prevent bacterial contamination	[204]
*T. o.* root extract	*E. coli*, *P. aeruginosa*, *S. aureus*, *S. typhi*, *B. subtilis*, *L. rhamnosum*	maximum values of the antibacterial properties were related to polysaccharides isolated from *T.* *o*. against four studied bacteria, and polysaccharides isolated, cyanidin-3-O-β-glucoside, N-acetylcysteine, and glutamic acid may play a major role	[206]

### 3.4. Hepatoprotective Activity

The liver, a vital organ responsible for detoxification, metabolism, and other essential functions, is susceptible to various diseases, including hepatitis, cholestasis, steatosis, granuloma, and vascular lesions, which can be caused by factors such as viral infections, excessive alcohol consumption, and exposure to toxic substances [207]. Additionally, factors such as drug use, air pollution, inflammation, elevated triglycerides, obesity, insulin resistance, elevated liver function, and elevated disorders. Chronic liver disease results from long-term liver damage, which reduces the organ’s ability to regulate blood sugar or remove toxins from the bloodstream [208]. Plants used in traditional medicine need a detailed investigation from an ethnopharmacological approach to treat liver disorders [209].

*Taraxacum* is commonly mentioned for its use in folk medicine to maintain liver health and to treat different liver ailments [13] because of its ability to “detoxify” blood. Dandelion is mentioned in folk medicine of China, India, and Russia as liver tonic [210], and dandelion leaves or roots are mentioned as being used for liver complaints in the Himalaya region [211]. Different traditional systems, including Ayurveda [212], Siddha, and Unani, recommend using dandelion for the management of liver disorders such as jaundiced liver and gallbladder disorders [213]. In the Chinese Pharmacopeia, the regular daily dose of dandelion is 10–15 g [214]. Additionally, the British Herbal Pharmacopoeia recommends a dose of 3–5 g of dandelion leaf as a choleretic agent, while its roots are used for hepatic function. For cholelithiasis or gallstone disease, 4–10 g of dried leaf or 2–8 g of dried root is recommended three times a day [215].

The protective effects of *Taraxacum* root against alcoholic liver damage were explored in vitro using HepG2/2E1 cells. In this study, reactive oxygen species were generated by ethanol treatment, which led to a reduction in cell viability by more than 40%. However, when cells were treated with both ethanol and *Taraxacum* hot water root extract, no cytotoxicity was observed compared to cells treated with ethanol alone. The aqueous extract from the root of *T. officinale* appears to act as an effective radical scavenger, targeting various radicals. Supplementation with *Taraxacum* root extract mitigated ethanol-induced liver damage, as demonstrated by the restoration of hepatic antioxidant levels and reduced lipid peroxidation. This suggests that the antioxidant-preserving properties of the *T. officinale* extract may play a crucial role in alleviating oxidative stress associated with ethanol toxicity [132].

The *T. officinale* extract has been studied for its potential effects on drug- and chemical-induced hepatic fibrosis in experimental animals, yielding promising results. For example, a water-ethanolic extract of dandelion root has been reported to alleviate CCl4-induced hepatic fibrosis in mice. Administering 600 mg/kg of dandelion extract for 10 days led to significant improvements in liver function, as indicated by the restoration of aspartate aminotransferase (AST), alanine aminotransferase (ALT), markers of hepatotoxicity, as well as increased levels of superoxide dismutase, hydroxyproline, and α-smooth muscle actin (α-SMA) protein expression [216]. Ovadje et al. highlighted the potential of natural products, particularly *Taraxacum* root extract, as non-toxic and effective alternatives to conventional chemotherapy treatments [217]. Additionally, *Taraxacum* extract was found to significantly promote the secretion of TNF-α and IL-1α and induce apoptosis in HepG2 cells [218].

Nazari et al. [219] conducted a study on the hepatoprotective effects of *Taraxacum syriacum boiss* root extract (TSBE) in rats. They found that TSBE significantly reduced liver damage caused by acetaminophen (APAP) toxicity. This was evidenced by decreased levels of liver enzymes (ALT, AST, and ALP = alkaline phosphatase) in the blood and reduced histopathological changes in the liver. Additionally, TSBE was shown to counteract APAP-induced oxidative stress by increasing antioxidant enzymes like catalase and glutathione levels. The antioxidant properties of TSBE are attributed to its phenolic compounds, which are capable of neutralizing harmful free radicals.

Ethanolic extracts of *T. officinale* root and leaves exhibited significant hepatoprotective effects. Studies have demonstrated that these extracts can reduce the levels of liver enzymes like AST and ALT, which are indicators of liver damage. Additionally, they can mitigate oxidative stress and lipid peroxidation, which are major contributors to liver injury. The protective effects of dandelion roots and leaves are attributed to their rich content of phenolic compounds. These compounds act as powerful antioxidants, neutralizing harmful free radicals and protecting liver cells from damage [220]. Overall, dandelion extracts, especially those derived from the roots and leaves, offer promising potential for the prevention and treatment of liver diseases. However, further research is necessary to fully elucidate the underlying mechanisms and to optimize the dosage and formulation of dandelion-based therapies.

Dandelion has been used in traditional medicine for centuries to address various health issues, particularly liver disorders. Its use is widespread, with cultures in Mexico [221], Bolivia [222], and Canada [223] recognizing its potential benefits for liver and biliary health. Modern research has begun to unravel the scientific basis for these traditional uses. Key findings include a. lipid-lowering effect—dandelion extracts have been shown to reduce levels of triglycerides, cholesterol, and high- and low-density lipoproteins (HDL and LDL) in animal models, suggesting their potential to improve lipid profiles [224]; b. liver protection against liver damage induced by various toxins, including CCl4. This protection is attributed to its antioxidant properties and ability to reduce inflammation, as well as to the presence of sesquiterpene lactones in the plant, which have been shown to protect against acute liver injury induced by CCl4 [131]. Hot water extracts of dandelion have also demonstrated hepatoprotective effects against CCl4-induced liver inflammation [149] and fibrosis liver [216] in rats; c. antioxidant activity—the phenolic compounds report in dandelion contribute to its strong antioxidant activity, facilitating the neutralization of harmful free radicals and protect liver cells from oxidative stress [14]; d. anti-inflammatory effects—dandelion extracts have been shown to degrade inflammation in the liver, which can impact liver damage.

The result of Al-Malki et al. revealed that the efficacy of dandelion leaves water extract as a hepatoprotective agent on DNA integrity against the CCl4 hepatotoxic effect was significantly improved at the end of the 6th week of administration, and the genomic DNA became more solid as it is demonstrated on the agarose gel electrophoresis. This indicates that *T. officinale* had a time-dependent hepatoprotective effect against the hepatotoxic effect of CCl4 [209].

Dandelion leaf extract has also demonstrated hepatoprotective effects against liver damage induced by sodium dichromate in rats. Daily oral administration of a hot water extract of dandelion leaf (500 mg/kg for 30 days) significantly reduced blood and liver levels of total cholesterol, triglycerides, AST, ALT, LDH, and chromium. This study suggests that the hepatoprotective effects of dandelion may be mediated by the upregulation of hepatic antioxidant enzymes [225]. Three cytotypes (2×), (3×), and (4×) of *T. officinale* from India were tested for their hepatoprotective properties: it was revealed that reduction in the activity of marker enzymes (AST, ALT) and the restoration of total protein, bilirubin levels, and liver weight; the hepatoprotective activity was the greatest in the diploid cytotype, which had found to have the highest concentration of cichoric acid (1.20% *w*/*w*), followed by the triploid (0.21 *w*/*w*) and tetraploid cytotypes (0.14% *w*/*w*) [199].

Lin et al. (2023) explored the protective effects of taraxasterol against liver damage caused by acetaminophen. Their in vitro experiments revealed that taraxasterol alleviated oxidative stress and inflammation in liver cells exposed to APAP. Additionally, the compound boosted the activity of the Nrf2 signaling pathway, a key cellular defense mechanism [226] (see Table 6).

Enhancements in liver function are correlated with the restoration of the liver cells’ histopathology. The hepatoprotective properties of dandelion can be attributed to its diverse chemical composition, including polysaccharides, flavonoids, phenolic, tannins, ascorbic acids, taraxol, taraxerol, laevulin, inulin, and luteolin. The mechanisms behind these hepatoprotective effects of dandelion may vary. Notably, oligofructans, which serve as prebiotics found in dandelion, promote the proliferation of beneficial gut bacteria while inhibiting the secretion of lipopolysaccharide, and they may also stimulate factors related to adipose tissue and lipid addition in the body. Additionally, dandelion exhibits anti-obesity properties by inhibiting digestive enzymes and affecting lipid metabolism and the formation of fat cells. The lipogenesis effects of dandelion are connected to a decrease in body and liver inflammation, as well as enhancements in insulin sensitivity and antioxidant condition [13].

Banday et al. (2023) suggested that the link between diabetes and liver diseases is stronger than mere coincidence [227], likely due to underlying mechanisms. Kim et al. (2023) further emphasized the significant risk of diabetes in the progression of chronic liver disease [228]. Devarbhavi et al. (2023) proposed a categorization of these conditions into three groups: diabetes-induced liver disease, liver disease associated with diabetes, and diabetes and liver disease occurring simultaneously [229].

**Table 6 ijms-26-00450-t006:** Hepatoprotective activity of *T. officinale* = *T.o*; *T. syriacum* = *T.s*.

Vegetal Organ/Phytocompounds	Type of Experiment	Results	Reference
*T. o.* leaves water extract	carbon tetrachloride-induced liver damage in male Wister albino rats	increased enzyme activities (AST, ALT, and LDH) at the end of the 2nd, 4th, and 6th weeks of the study. By the 6th week of treatment, significant improvement and repair of genomic DNA were observed compared to the genomic DNA of untreated animals	[209]
*T. o.* root water extract	CCl_4_-induced hepatic fibrosis in mice	reduced the accumulation of hepatic fibrinous deposits, restored the histological structure, and regulated the expression of GFAP and α-SMA by inactivating hepatic stellate cells while also enhancing the liver’s regenerative capacity	[216]
*T. o.* extract	human hepatoma cell line, Hep G2	decreased the cell viability by 26% and significantly increased the (TNF)-α and IL-1α	[218]
*T. s.* Ethanol extract of root	male Wistar rats with hepatocellular injuries induced by acetaminophen (APAP)	decrease in serum levels of glutamate oxaloacetate transaminase (SGOT), serum glutamate pyruvate transaminase (SGPT), and ALPwas observed, accompanied by the prevention of histopathological alterations in the liver	[219]
*T. o.* leaves extract	APAP-induced hepatotoxicity	reduce thiobarbituric acid-reactive substance levels, prevent the reduction of sulfhydryl levels, and increase serum aspartate and alanine aminotransferase levels	[220]
*T. o.* ethanolic and n-hexane extract of leaves	CCl4-induced liver toxicity in rats	both leaf extracts reduced the concentrations of TBARS, H_2_O_2_, and nitrite, with the ethanolic leaf extract demonstrating a more effective protective effect.	[224]
Taraxasterol	APAP-induced acute liver injury in mice	protected liver cells from APAP-induced damage by reducing oxidative stress and inflammation and increasing Nrf2 expression.	[226]
Chicoric acid	High-fat diet-induced obese mice treated with chicoric acid	reduced obesity in high-fat diet-fed mice by alleviating insulin resistance, liver injury, and inflammation while boosting the antioxidant defense system	[230]

### 3.5. Anti-Diabetic Activity

Diabetes is a common and one of the most significant health problems worldwide: the International Diabetes Federation (IDF) Atlas (2021) projections show that 1 in 8 adults will be living with diabetes, an increase of 46% [231] Prolonged insulin resistance and pancreatic stress associated with diabetes can result in severe complications, including blindness, kidney failure, and heart disease [232]; it impacts critical organs like the liver, muscles, and adipose tissue [233], complications on which dandelion also registers a good influence. Oxidative stress, caused by self-oxidation and protein glycation, worsens glucose regulation [234]. This process leads to increased lipid peroxidation and diminished antioxidant defenses, contributing to ß-cell dysfunction [235,236]. Additionally, glucotoxicity and lipotoxicity impair insulin secretion by disrupting the conversion of proinsulin into insulin [237].

Research suggests that dandelion extracts may stimulate insulin release from pancreatic β-cells, helping to counteract high blood sugar levels [238,239]. Plant-based products and compounds have shown promise in managing diabetes through various mechanisms, including inhibiting sugar-digesting enzymes, reducing glucose reabsorption in the kidneys, and influencing glucose metabolism in the liver. Dandelion’s antidiabetic properties are attributed to its bioactive compounds, such as phenols, flavonoids, phenolic acids, chicoric acid, taraxasterol, chlorogenic acid, and sesquiterpene lactones [7,235].

Different dandelion root extracts, including aqueous, methanolic, chloroform, ethyl acetate, acetone, and petroleum ester, have been evaluated for their antidiabetic effects in both normal and diabetic mice.

α-glucosidase, an enzyme that breaks down complex carbohydrates into simple sugars, plays a role in postprandial blood sugar spikes. By inhibiting this enzyme, dandelion extracts can slow down the absorption of glucose from the digestive tract. It was found that dandelion extracts inhibit α-glucosidase [240]. Additionally, the polyphenols and flavonoids in dandelion products may help regulate gene expression, which is implicated in lipid accumulation, oxidative stress, and insulin resistance [241].

The dandelion root is rich in inulin, a type of fructo-oligosaccharide, which may assist in regulating blood sugar levels. Mir et al. observed that high concentrations of its water extract can decrease hyperglycemia. Their research indicated that both methanol and water extracts of *T. officinale* show substantial inhibitory effects on the enzymes α-amylase and α-glucosidase, with water extracts from all parts of the plant (roots, flowers, and stems) displaying slightly stronger inhibition than methanol extracts. This difference may be attributed to a higher presence of ionic components in water extracts. The study revealed a dose-dependent increase in inhibitory activity against the α-amylase enzyme, showing a more pronounced effect compared to α-glucosidase [242]. Li and colleagues concluded that the aqueous extract of dandelion root, containing 63.92 ± 1.82 mg/g of polysaccharides, 2.57 ± 0.06 mg/g of total flavonoids, 8.93 ± 0.34 mg/g of total phenolic compounds, and 0.54 ± 0.05 mg/g of saponins, exhibited a statistically significant ability to inhibit the activities of both α-glucosidase and α-amylase [243].

The research conducted by Frolova et al. indicated that both quercetin and vitamin C significantly contribute to the antioxidant properties of *T. officinale* root extract. This extract is notably rich in B vitamins, specifically pyridoxine (156.40 μg/mL) and thiamine (46.20 μg/mL), and also contains flavonoids like rutin and quercetin [244]. In an in vivo study, the administration of a 400 mg/kg aqueous extract of dandelion root resulted in a substantial reduction in blood glucose levels (62.33%) in alloxan-induced diabetic mice. In contrast, other extracts did not exhibit significant effects in this model, and no changes in glycemia were observed in non-diabetic mice. Phytochemical analysis revealed a higher total phenolic content than flavonoids in the aqueous extract, with identified compounds including chlorogenic acid, protocatechuic acid, and luteolin-7-glucoside [245]. Additionally, Zolotova et al. demonstrated that the polysaccharide content in dandelion roots surpassed that of *Arctium lappa*, though the latter exhibited greater total phenolic content, tannin levels, DPPH assay results, and α-amylase activity. Qualitative analyses confirmed the presence of inulin in both dandelion and burdock roots [246].

Nnamdi et al. have studied the effects of aqueous and ethanolic extracts of *T. officinale* leaves and roots on fasting blood glucose (FBG) levels in both normal and streptozotocin-induced diabetic Wistar albino rats. The administration of *T. officinale* leaf and root extracts (6% and 10% concentration) for 21 days determined a statistically significant decrease in glucose concentration. This study strongly suggests that although *T. officinale* leaves and roots possess hypoglycemic properties, the roots of ethanolic extraction are relatively more potent (see Table 7) and may be beneficial in the management of diabetes [247]. Leafy extracts have anti-oxidative activities expressed by enzymes, including glutathione peroxidase, glutathione reductase, and superoxide dismutase [248]. Dandelion leaf extract possesses anti-inflammatory effects that may help shield against acute pancreatitis induced by cholecystokinin in rats. The extract of *Taraxacum*, which contains terpenoid and bitter sterol compounds like taraxacin and taraxacerin, has been utilized to address issues related to the liver and gallbladder [159].

Dandelion flowers are potential antioxidant resources, exerting their effect through the rich content of phenolic components, including flavonoids, coumaric acid, and ascorbic acid [235]. Ethyl acetate fraction of dandelion flowers, with bioactive components such as luteolin and luteolin 7-O-glucoside, has scavenged ROS by preventing DNA from ROS-induced damage [249]; Tfayli’s research explored the antioxidative effects of dandelion, revealing that a reduction in glucose transport into muscle cells leads to elevated glucose and fat levels in the blood plasma, ultimately resulting in hyperglycemia and lipid oxidation. This phenomenon is linked to β-cell dysfunction, where the pancreas fails to produce sufficient insulin, a condition associated with increased free fatty acids in the plasma [250]. Oxidative stress contributes to cellular damage, affecting key components like DNA, lipids, and proteins. Over time, this damage can lead to metabolic disorders, a critical factor in the development of type 2 diabetes [251].

**Table 7 ijms-26-00450-t007:** Anti-diabetic activity of *T. officinale* = *T.o*; *Momordica charantia* = *M.c*.

Vegetal Organ/Phytocompounds	Type of Experiment	Results	Reference
*T. o*. methanol extract of stem, flowers, and roots	alpha-amylase and alpha-glucoside inhibition	stem exhibited the strongest antidiabetic activity, followed by the roots, while the flowers were the least effective	[242]
*T. o*. root water, methanol, Ethanol, n-hexane, ethyl acetate, and chloroform extract	IR-HepG2 cells were grown in complete medium DMEM	water extract of dandelion, rich in polysaccharides, total flavonoids, phenolic compounds, and saponins, showed significant inhibitory effects on α-glucosidase and α-amylase while enhancing hexokinase and pyruvate kinase activity	[243]
*T. o.* roots ethanol extract	chemical compounds in dandelion and burdock roots	liquid chromatography-mass spectrometry was employed to tentatively identify chemical components. Qualitative analysis confirmed the presence of inulin in the root, with higher tannin content and α-amylase activity observed in burdock compared to dandelion	[244]
*T. o.* root water extract	normoglycemic and diabetic mice evaluated at two dosages (200 mg/kg and 400 mg/kg) using antidiabetic tests and subcutaneous glucose tolerance assessments	400 mg/kg extract effectively lowered blood glucose levels, while the aqueous extract significantly enhanced glucose uptake	[245]
*T. o.* root Ethanol extract	hypoglycemic properties of the extracts based on α-amylase activity	plant extract did not match the efficacy of acarbose, and its suitability as an antidiabetic agent remains uncertain without further in vivo studies	[246]
*T. o.* ethanolic and aqueous extract of leaves and roots	normal and streptozotocin-induced diabetic Wistar albino rats *(Rattus rattus*) were studied	extracts at 6% and 10% concentrations reduced fasting blood glucose levels, with the ethanolic root extract showing relatively higher potency	[247]
Chicoric acid	high-fat diet-induced obese C57BL/6J mice treated with chicoric acid	extract alleviated insulin resistance, liver damage, and inflammation in mice	[230]
*T. o.* and *M. c.* ethanol extracts	streptozotocin-nicotinamide induced diabetic rats—male Wistar albino rats and male Sprague Dawley rats	polyherbal combination demonstrated enhanced antidiabetic effects, including enzyme inhibition and blood glucose reduction, comparable to standard treatments like Glibenclamide and Metformin.	[252]

It was demonstrated that chlorogenic acid impacts insulin secretion and sensitivity [230], probably through its potent antioxidant activity, which may suppress oxidative stress markers such as malondialdehyde and glutathione [253]. Chicory acid increases glucose uptake in muscle cells due to the stimulation of insulin secretion in the pancreas [254]. Schütz et al. demonstrated that chicory acid and taraxasterol inhibit α-glucosidase and α-amylase, preventing the digestion of complex carbohydrates such as starch and thus contributing to the anti-hyperglycemic effect [14].

Taraxasterol may improve β-cell function by stimulating insulin gene expression and inhibiting cell degradation [255]. Polyphenols in dandelion can enhance insulin secretion by stimulating cAMP signaling, preventing oxidative stress, and promoting β-cell regeneration and growth [256]. Other bioactive compounds in dandelions, such as alkaloids, glycosides, terpenoids, amino acids, inorganic ions, steroids, and galactomannan gum, all of which have been shown to influence glucose uptake and metabolism [257].

The analysis of phytochemicals in dandelion fruits has also been studied. Lis et al. [258] prepared a methanolic extract from dandelion fruit and identified the presence of hydroxycinnamic acid derivatives and flavone derivatives. Multiple parameters were examined, including lipid peroxidation, protein carbonylation, thiol oxidation, and platelet adhesion. Their findings indicated that the extracts of hydroxycinnamic acid, flavonoids, and the luteolin fraction demonstrated significant antioxidant and antiplatelet activities.

Perumal et al. (2023) demonstrated that a moderate concentration of the polyherbal combination exhibited hypoglycemic effects in diabetic rats [252]. The combination of *T*. *officinale* and *Momordica charantia* extracts was found to be more effective than individual extracts. In addition to their direct antihyperglycemic effects, these polyherbal combinations may also indirectly contribute to blood sugar control through antidiuretic and antilipolytic activities. Dandelion, with its antihyperglycemic, antioxidant, and anti-inflammatory properties [259], is recognized as a valuable medicinal plant. This potential is attributed to the various bioactive compounds present in dandelion, which offer promise as both pharmaceuticals and nutraceuticals for diabetes management.

### 3.6. Immunomodulatory Action

Dandelion polysaccharides are frequently mentioned for their potential immunomodulatory properties. Monocytes and macrophages play a crucial role in inflammatory responses, acting as primary sources of pro-inflammatory mediators and enzymes. These include TNF-α, various interleukins (ILs), (COX), and (NOS). TNF-α and IL-1β function as signaling molecules for immune cells, orchestrating the inflammatory response. (COX-2) is a fundamental enzyme for the synthesis of pro-inflammatory prostaglandins, making it a common target for many anti-inflammatory therapeutic plants. Nitric oxide (NO), a free radical, plays a role in numerous physiological and pathophysiological processes, including neurotransmission and inflammation. The inducible form of iNOS in activated macrophages is primarily responsible for generating elevated concentrations of NO during inflammatory events. Consequently, inhibiting the expression of iNOS has become a significant objective for numerous anti-inflammatory medicinal plants to exert their effects [260].

The immunostimulatory action induced by *Taraxacum* root extract supplements may be attributed to several cellular and humoral antibody-mediated immune mechanisms against anthelmintic infections. The extract also improves the therapeutic effects of Praziquantel tablets on liver and intestinal disorders in mice infected with *Schistosoma mansoni, which* have been shown to enhance antioxidant, anti-fibrotic, and immunomodulatory responses. Treatment with dandelion resulted in significant morphological alterations of male adult worms, a marked reduction in both ova count and worm burden, as well as notable histological changes in the liver and intestines. This included a substantial decrease in the diameter of fibrinous granulomas, positive effects on cellular immune responses, oxidative stress markers (malondialdehyde andNO), and the production of interleukin-10. Additionally, there was an upregulation of total immunoglobulin G, interferon-gamma, and antioxidant enzymes such as superoxide dismutase and catalase compared to untreated infected mice [261]. Zhou et al. [147] demonstrated that dandelion polysaccharides effectively lowered serum malondialdehyde levels while enhancing the activities of superoxide dismutase and glutathione peroxidase. Moreover, these polysaccharides diminished the expression of the chemotactic factor MCP-1 and pro-inflammatory cytokines (TNF-α, IL-1β, and IL-6) in atherosclerotic lesions. Furthermore, dandelion has been observed to alleviate inflammation by suppressing the production of pro-inflammatory cytokines, thereby helping to reduce tissue damage and support healing processes.

Shekarabi et al. (2023) investigated the impact of dietary dandelion flower extract (DFE) on rainbow trout’s immune response and disease resistance [262]. They found that DFE supplementation significantly reduced mortality rates after bacterial infection. *T. mongolicum* polysaccharide (TMP) has shown promise as an anti-inflammatory and antioxidant agent in other studies. When added to fish feed, TMP can improve growth, digestion, immune function, and antioxidant status, as well as regulate gene expression. A comparative analysis of dandelion root and leaf extracts from the Dobrogea region revealed that spring-harvested plants exhibited slightly higher antioxidant activity compared to autumn-harvested plants [263].

Taraxasterol isolated from the traditional Chinese *Taraxacum* [133] significantly suppressed paw swelling and arthritis index, decreased the spleen index and thymus index, and attenuated body weight loss in studying the antiarthritic effect of taraxasterol on arthritis induced by Freund’s complete adjuvant in rats. It also inhibited the overproduction of serum TNF-α, IL-1β, and PGE2 and increased serum OPG production in rats. Histopathological examination revealed attenuated synovial hyperplasia and diminished bone and cartilage. In vitro cell and in vivo animal studies support the beneficial effects of dietary flavonoids on glucose homeostasis, because flavonoids regulate carbohydrate digestion, adipose deposition, insulin release, and glucose uptake in insulin-responsive tissues [264]. Dandelion root polysaccharides exhibit an ameliorative effect on ulcerative colitis induced by dextran sodium sulfate [265]. In vitro studies demonstrated that *T. coreanum* chloroform (TCC) fraction significantly reduced inflammation by inhibiting the production of NO and prostaglandin E2 (PGE2), as well as the expression of iNOS and COX-2 enzymes. Additionally, TCC suppressed the activation of the NF-κB signaling pathway. In a mouse model of septic shock, TCC treatment decreased the levels of TNF-α, IL-1β, and IL-6 [266] (see Figure 7) and significantly increased survival rates.

Yoon (2024) investigated the impact of hot-water and cold-water dandelion root extracts [267] on the immune response in mice. Both extracts were found to enhance the production of pro-inflammatory cytokines, such as IL-6 and IL-12, by macrophages, suggesting a potential modulation of the innate immune response (see Table 8).

Ethanol extracts from *T. officinale* have demonstrated anti-angiogenic, anti-inflammatory, and pain-relieving properties. These extracts help reduce the production of NO and COX-2, as well as decrease the levels of ROS in activated macrophages. Additionally, they can slow down the expression of iNOS and COX-2 in LPS-stimulated RAW264.7 cells [161]. Given that extracts from anti-inflammatory medicinal plants typically contain multi-components, it is anticipated that they target various pathways to influence the intricate balance of cellular immune networks. This multi-target approach may be more beneficial compared to agents that focus on a single target, as successfully altering inflammatory disease processes likely requires coordinated actions across several genes. Consequently, the effectiveness of herbal therapies and compounds derived from plants may rely on their ability to impact more than one target [260].

### 3.7. Antitumoral Activity

Dandelion has been used in traditional ethno-medicinal systems (i.e., Chinese, Arabian, Indian, and Native American) to treat divergent types of cancer [268], and in the last 10 years, many studies have shown that *Taraxacum* exerts antitumoral effects—breast, gastric, and liver cancer, neuroblastoma, lung carcinoma, and prostate cancer.

Both in vivo and in vitro investigations have shown that extracts from this plant’s roots can trigger programmed cell death in various types of cancer cells, including human leukemia, colorectal, and pancreatic cancer cells [269]. Additionally, these extracts inhibit the proliferation and growth of breast cancer cells by modulating the phosphatidylinositol 3-kinase (PI3K)/AKT signaling pathway [270], as well as affecting prostate cancer cells [269,271]. Research conducted by Zhu et al. demonstrated that the root extracts of *Taraxacum* resulted in the downregulation of the long non-coding RNA colon cancer-associated transcript-1 (CCAT1) in gastric cancer cells treated with dandelion. This reduction in CCAT1 expression subsequently inhibited both the proliferation and migration of gastric cancer cells, all while not causing toxicity in non-cancerous cells [272].

In neuroblastoma cell lines (SH-SY5Y and Kelly), *Taraxacum* caused apoptosis and loss of mitochondrial integrity as well as inhibition of invasion and migration. Aqueous fermented *T. officinale* extract on this pediatric cancer cell line was more effective than the normal human fibroblast cell line, NHDF-C [273].

*Taraxacum* has been known to effectively block the inflammation and cancer development process by intervening PI3K/Akt, NF-κB and apoptosis signaling pathways, coupled with the downregulation of lncRNA, improvement of cell microenvironment, reduction of inflammatory factor secretion, promotion of fatty acid degradation and the improvement of intestinal flora [274]. Flavonoids from *Taraxacum* have been demonstrated to inhibit tumor cell proliferation via inhibition of ROS formation, as well as suppression of xanthine oxidase, COX-2, and 5-LOX, which are the major catalysts for tumor promotion and progression [275]. Other studies proved that cyclin-dependent kinases (CDKs) are key regulators of cell cycle progression, immune cell activation, neo-angiogenesis, and inflammation [276].

El-Emam et al. [78] purified from *Taraxacum* leaves a bioactive compound, β-branched glucomannan, which demonstrates a strong IC_50_ against non-small cell lung carcinoma (NSCLC). To enhance the properties of the extract, researchers developed nano-functional particles using various nanocarriers, including liposomes and solid lipid nanoparticles. The findings indicated that β-branched glucomannan not only promoted apoptosis but also caused cell cycle arrest during the S-phase and G2/M phase, accompanied by an increase in Beclin-1 and Bax levels, along with a reduction in Bcl-2 expression.

In a study by Lin et al., aqueous extracts from two species of *Taraxacum*, *T. mongolicum* and *T. formosanum*, were evaluated for their antitumor effects against three human breast cancer cell lines: MDA-MB-231, ZR-75-1, and MCF-7. Both extracts inhibited cell migration and colony formation across all three cell lines and showed suppressive effects on the proliferation of MCF-7 cells by triggering apoptosis, decreasing cell growth, disrupting mitochondrial membrane potential, and/or reducing oxygen consumption rates. The study found that T. mongolicum exhibited higher cytotoxicity against all three breast cancer cell lines, particularly targeting the triple-negative breast cancer cell line MDA-MB-231. Conversely, *T. formosanum* induced ribotoxic stress in both MDA-MB-231 and ZR-75-1 cells, while *T. mongolicum* did not [277].

Taraxerol, a triterpene compound from the plant *Taraxacum*, has shown promise as a potential anticancer agent. Studies have demonstrated its ability to inhibit cell metastasis, induce apoptosis, and suppress cell proliferation in various cancer cell lines. For instance, Taraxerol has been shown to target the Hippo and Wnt signaling pathways in gastric cancer cells [278], the mitochondrial pathway in cervical cancer cells [279], and the Akt pathway in bladder cancer cells [280]. Recent research by Xia et al. (2023) further explored the effects of taraxerol on breast cancer cells [281]. They found that taraxerol significantly reduced cell migration and invasion by inhibiting the phosphorylation of ERK.

Fulga et al. [282] investigated the antitumor properties of *T. officinale* leaf extracts against U-138 MG glial cells. They found that extracts prepared with dimethyl sulfoxide (DMSO), 50% ethanol, and 80% ethanol exhibited significant antitumor activity (see Table 9). This effect was correlated with the concentration of hydroxycinnamic acids, specifically chicoric, chlorogenic, and caftaric acids, present in these extracts. The concentration ranges of these acids in the active extracts were as follows: chicoric acid (904–52,500 µg/mL), chlorogenic (114.4–1746 µg/mL), caftaric (70.4–8460 µg/mL).

Wang et al. (2023) demonstrated that taraxasterol, at a concentration of 10 µg/mL, significantly increased the viability of bovine mammary epithelial cells [283]. The study revealed that taraxasterol reduced oxidative stress by decreasing the accumulation of ROS. Furthermore, it alleviated endoplasmic reticulum (ER) stress by downregulating the expression of glucose-regulated protein 78 kDa (GRP78) and activating transcription factors 6 and 4. Additionally, taraxasterol was found to reduce cell apoptosis by suppressing the expression of caspase-3 and Bcl-2-associated X (BAX) and upregulating the expression of the anti-apoptotic protein B-cell lymphoma-2 (Bcl-2).

The effect of different compounds present in *T. officinale* in distinct types of cancer in laboratory conditions might lead to the discovery of novel anticancer agents by Tiwari et al. [284]. Aras et al. [285] investigated the cytotoxic effects of ethanol extract obtained from fresh latex-filled dandelion roots (DR) on human pancreatic cancer cells and demonstrated that treatment of PANC-1 cells with 10 mg/mL of DR extract produced the maximum inhibition rate, and the lowest IC_50_ value was reached during the 72-h treatment period. Hepatocellular carcinoma (HCC) is one of the most common malignancies, which accounts for 90% of primary liver cancer. HCC usually presents with poor outcomes due to the high rates of tumor recurrence and widespread metastasis [286]. A methanolic extract of dandelion root was observed to decrease the growth of apoptotic cells (HepG2 cells) line and enhance the phosphorylation level of AMPK of HepG2 cells, which is considered crucial in cancer treatment and other metabolic diseases. Rehman et al.’s research clearly demonstrated the potency of dandelion root extract against liver cancer [268].

Gui and Fan (2023) assessed the impact of dandelion flavone on multiple myeloma (MM) cells using the CCK-8 method. Their study revealed that taraxacine flavonoids inhibited the proliferation, migration, and invasion of MM cells and induced apoptosis by targeting the PI3K/AKT pathway [287]. This suggests that the leaves and roots of *T. officinale*, which are rich in these bioactive compounds, may have significant anticancer potential, particularly against MM.

**Table 9 ijms-26-00450-t009:** Antitumor activity of *T. officinale* = *T.o*; *T. formosanum* = *T.f*; *T. mongolicum* = *T.m*; *T. campylodes* = *T.c*.

Vegetal Organ/Phytocompounds	Type of Experiment	Results	Reference
*T. o.* taraxasterol	concanavalin A-induced acute hepatic injury in mice	inhibiting T toll-like receptors/NF-κB (-) inflammatory signaling pathway and promoting Bax/Bc1-2 anti-apoptotic signaling pathway	[171]
*T. o.* water root extract	BxPC-3 and PANC-1 pancreatic cells	induce apoptosis and autophagy in human pancreatic cancer cells with no significant effect on noncancerous cells	[217]
*T. c.* aerial part chloroform fraction	mouse peritoneal macrophages stimulated in vitro with interferon-γ and lipopolysaccharide in a mouse model of lethal septic shock	inhibited the production of (TNF)-αIL-1β, and IL-6, and increased survival by 83%	[266]
*T. o.* methanolic extract of roots	HepG2, MCF7, HCT116, and normal Hs27 cell lines	500 µg/mL decreased the growth of the HepG2 cell line, while the effect on MCF7 and HCT116 cell lines was less pronounced, and no effect has been observed in Hs27 cell lines; enhanced the phosphorylation level of AMPK of HepG2 cells, which is considered crucial in cancer treatment	[268]
*T. o.* water extract of roots	prostate cancer cell line DU-145 cultured in Eagle’s Minimum Essential Medium	exhibit selective anticancer activity, and in addition to the chemotherapeutics, taxol, and mitoxantrone were determined to enhance the induction of apoptosis, significantly reduce the tumor burden in prostate cancer xenograft models	[269]
*T. o.* water extract of roots	female albino rats with breast cancer	regulated PI3K and Akt pathways involved in the suppression of breast cancer growth and proliferation	[270]
*T. o.* water extract of roots, leaves and flowers	MCF-7/AZ breast cancer cells and LNCaP prostate cancer cells	*T. o.* root extract blocks the invasion of MCF-7/AZ cells, but leaf extract blocked the invasion of LNCaP cells into collagen type I and diminishes the expansion of MCF-7/AZ cells	[271]
*T. o.* water extract of roots	gastric cancer (GS) cell lines (SGC7901, BGC823), and a normal gastric epithelium cell line (GES-1)	decrease the expression of the long non-coding RNA colon cancer-associated transcript-1 in gastric cancer cells, which is associated with reduced cell proliferation and migration	[272]
*T. o.* water extract of plant	neuroblastoma cell lines, SH-SY5Y and Kelly	caused apoptosis and loss of mitochondrial integrity, as well as an inhibition of invasion and migration	[273]
*T. m.*, *T. f.* water extract	MDA-MB-231 human triple-negative breast cancer cells, as well as ZR75-1 and MCF-7 non-triple-negative breast cancer cells	cytotoxic effects against all three breast cancer cell lines were observed, particularly in MDA-MB-231 cells. The cytotoxicity was mediated through apoptosis, reduced cell proliferation, and disruption of the mitochondrial membrane potential	[277]
*T. o.* taraxerol	MDA-MB-231 human breast cancer cell	inhibited the migration and invasion of cells via the ERK/Slug axis	[281]
*T. o.* ethanol and dimethyl sulfoxide extract of leaves	human glioblastoma cell lines U-138 MG DMEM medium	*T. o.* extracts were compared with doxorubicin and were close to that. The best antitumor activity was shown by *T. o.* extracts prepared with DMSO (110,000 µg/L—17.3 ± 8%)	[282]
*T. o.* ethanol extract root	PANC-1 (human epithelioid carcinoma) cell in DMEM medium	10 mg/mL of DRE produced the maximum inhibition rate, and the lowest IC_50_ value was reached during the 72-h treatment	[285]
*T. o.* flavone	human MM = Multiple myeloma cell line U266	reduced the expression of matrix metalloproteinases (MMP-2 and MMP-9) while increasing the expression of tissue inhibitors of metalloproteinases (TIMP-1 and TIMP-2). Additionally, it promoted the expression of pro-apoptotic proteins and inhibited the polarization of macrophages towards the M2 phenotype by suppressing the PI3K/AKT signaling pathway	[286]

### 3.8. Antioxidant Activity

Excessive levels of (ROS), often termed oxidative stress, can lead to damage of lipids and proteins within the blood’s clotting system, potentially increasing its activity [288]. This heightened ROS production is linked to protein oxidation and subsequent inflammatory responses [289,290]. Furthermore, excessive ROS can induce tissue damage, triggering inflammation [291,292,293]. Polyphenols, renowned for their antioxidant properties, can mitigate inflammation by disrupting the ROS-inflammation cycle. They achieve this by neutralizing a broad spectrum of ROS. Additionally, polyphenols can directly scavenge free radicals and bind metal ions, such as quercetin’s ability to chelate iron [294]. Following the work of Mishra et al., polyphenols have been shown to inhibit various enzymes involved in ROS generation [295].

Hagymasi et al. demonstrated that extracts from dandelion leaf and root are hydrogen-donating, ROS formation-inhibiting, and radical-scavenging [296] (see Figure 8). Antioxidant activities of dandelion polysaccharides also prevented the metamorphism of fresh white shrimps [297]. A study conducted by Khan et al. [298] revealed that hydro-alcoholic extracts exhibited better antioxidant activity as compared to aqueous extracts. The phenolic contents of *T. officinale* portray the fact that extract from this herbal plant may help discover new antibiotic substances for chemotherapy and control of chronic infectious diseases.

Yoon and Park assessed the antioxidant properties of water and ethanol extracts from *T. officinale* (TOWE and TOEE, respectively) by examining their ability to combat oxidative stress and induce heme oxygenase-1 (HO-1) in RAW 264.7 cells [299]. Antioxidant activity was evaluated through radical scavenging assays, cell protection against oxidative damage, and Western blot analysis. The role of HO-1 induction in the antioxidant effects of both extracts was confirmed using the selective HO-1 inducer, cobalt protoporphyrin (CoPP), and inhibitor, tin protoporphyrin (SnPP). These findings suggest that TOWE and TOEE effectively mitigate oxidative damage by activating the Nrf2/MAPK/PI3K signaling pathway, leading to increased HO-1 expression in RAW 264.7 cells.

Jedrejek et al. have shown that dandelion roots are a safe and potent source of diverse natural compounds with antioxidant, anticoagulant, and antiplatelet properties. These roots contain a variety of bioactive substances, including hydroxycinnamic acids (HCAs), sesquiterpene lactones (SLs), and compounds like 4-hydroxyphenylacetate inositol esters (PIEs). Through detailed LC-MS analysis, they identified approximately 100 phytochemicals, some of which were previously unknown in both the genus *Taraxacum* and the plant kingdom, such as amino acid-SL adducts [300]. Among the various dandelion root preparations tested, those enriched in SL-amino acid adducts (preparation A) and PIEs (preparation C) exhibited the strongest protective effects against oxidative damage induced by H_2_O_2_/Fe in blood plasma. Preparations enriched with HCAs demonstrated significant anticoagulant activity. Importantly, none of the dandelion root preparations caused platelet lysis at concentrations ranging from 0.5 to 50 μg/mL.

The flavonoids present in dandelion extract have been shown to positively impact cardiovascular health by exerting antioxidant effects and upregulating the expression and activity of antioxidant enzymes [301,302]. A study by Radoman et al. [303] in 2023 demonstrated that rats treated for four weeks with dandelion root freshly boiled in 250 mL water had a statistically significant increase in hydrogen peroxide (H_2_O_2_) and in lipid peroxidation index values compared to the control group that drank tap water and also superoxide anion radical (O^2–^) values were statistically significantly lower (*p* < 0.05) in the rats treated with dandelion water than the control group. In the same year, Wang et al. [72] made a groundbreaking discovery by identifying hesperetin-5′-O-β-rhamnoglucoside as a novel flavonoid in *T. mongolicum* Hand.-Mazz. Their subsequent evaluation confirmed the potent antioxidant activity of this compound, primarily attributed to the synergistic effect of various substituents on the B ring and its enhanced hydrogen-donating capacity.

Yi et al. [175] elucidated the structural requirements for 4-oxo-flavonoids to protect against oxLDL-induced endothelial dysfunction, highlighting the importance of 3′,4′-ortho-dihydroxyl, 3-hydroxyl, 2,3-double bond, and 5,7-meta-dihydroxyl groups.

Xue et al. [304] employed ultrasound-assisted hot water extraction to isolate polysaccharides from *T. mongolicum* (TPMs). Through a purification process involving DEAE-52 and Sephadex G-100 chromatography, they obtained a homogeneous polysaccharide fraction (TMPs-1-SG) (see Table 10). In vitro antioxidant assays revealed that TMPs-1-SG effectively scavenged DPPH and OH radicals, with IC_50_ values of 0.71 mg/mL and 0.75 mg/mL, respectively.

The in vivo anti-inflammatory study in rat turpentine-induced inflammation showed, by the oxidative stress determinations, that the *T. officinale* tincture decreases the total oxidative stress, the oxidative stress index, and the total antioxidant capacity. Dandelion tincture significantly attenuated levels of malondialdehyde, thiols, and nitrites/nitrates, indicative of reduced oxidative stress, in both inflammatory and isoprenaline-induced myocardial infarction rat models. Furthermore, the tincture significantly decreased levels of aspartate aminotransferase, alanine aminotransferase, creatine kinase-MB, and (NF-κB), markers of inflammation and cardiac injury [172].

Wójciak et al. (2024) demonstrated that the addition of dandelion leaf extract (0.5–1%) significantly improved the oxidative stability of raw-ripening sausages with a reduced nitrite content (80 mg/kg). This improvement was attributed to the extract’s strong antiradical activity against ABTS+ and DPPH radicals, as well as its ability to reduce lipid peroxidation, as measured by the TBARS index [305]. Wu et al. (2024) obtained dandelion polysaccharides (SDRPs) from crude dandelion roots through hot-water extraction and sulfation. Their results suggest that SDRPs have a positive effect on the growth of beneficial gut bacteria, specifically *Lactobacillus plantarum* and *Lactobacillus acidophilus*, indicating their potential as a prebiotic [306].

The anti-inflammatory and antioxidant properties of dandelion make it a promising natural remedy for various inflammatory conditions. Further research is needed to fully understand the mechanisms underlying its therapeutic effects and to develop standardized extracts and formulations for clinical use.

## 4. Potential Toxicity of *Taraxacum* Species

Dandelion, with its centuries-long history of safe consumption as both a food source and herbal medicine, has a well-established safety profile. Numerous animal studies have confirmed the low toxicity of both fresh and processed dandelion. Dandelion root and dandelion extracts have a “generally recognized as safe” status and have been approved by the FDA for use in dietary supplements. While occasional mild side effects, such as diarrhea, upset stomach, allergies, or skin irritation, may occur in some individuals, serious adverse effects are rare.

Previous research has primarily focused on isolating and identifying specific bioactive compounds from various dandelion species. However, in recent years, there has been a growing interest in studying the biological activities of partial extracts, such as root and leaf extracts, as well as investigating the potential of other active ingredients within the plant. While in vitro cell culture experiments and in vivo animal models have provided valuable insights into the pharmacological mechanisms of dandelion’s active ingredients; there remains a need for further clinical research to translate these findings into effective therapies.

The diverse phytochemical composition of *T. officinale* offers significant potential for the development of novel therapeutic agents. By leveraging advanced technologies such as bioinformatics, nanotechnology, genetics, and biotechnology, researchers can further explore the isolation, purification, and optimization of these bioactive compounds. This interdisciplinary approach can accelerate the discovery and development of innovative drug therapies derived from natural sources.

## 5. Conclusions

The present review provides evidence that dandelion flowers, stems, roots, and leaves are rich in various bioactive compounds. Dandelion crude extracts and purified compounds have shown different biological activities. Published studies revealed support for antitumor, anti-inflammatory, antioxidant, antimicrobial, and other pharmacological activities.

Roots, leaves, and flowers of dandelion were tested for their anti-inflammatory activity in vitro and in vivo, revealing that taraxasterol, chicoric acid, polysaccharides, and luteolin can effectively inhibit the inflammatory reaction; our research found that the most frequent vegetal organ used was the leaf. Alcoholic, hot-water, and cold-water extracts of dandelion vegetal organs or the whole plant revealed an important impact in modifying inflammatory disease processes.

Aqueous, alcoholic, and organic extracts of different plant parts: root, leaf, and flower demonstrated antiviral and Gram-positive and Gram-negative antibacterial properties and relevant antifungal activity. Antiviral activity was especially demonstrated by studies on leaf extract (water and methanol), and antibacterial properties were revealed on roots and leaves extracts against *Staphylococcus aureus*, *Enterococcus faecalis*, *Escherichia coli*, *Bacillus subtilis*, etc.

*Taraxacum* root and leaves infusions or extracts significantly attenuated marker enzymes of liver toxicity, demonstrating hepatoprotective activity and improving liver function due to its content of taraxasterol, taraxerol, polysaccharides, inulin and luteolin, tannins, and ascorbic acid.

The immunostimulatory action induced by *Taraxacum* root extracts was frequently investigated via NO synthesis inhibition, COX-2 expression, and PGE2 by luteolin and luteo-lin-7-O-glucoside or through the release of inflammatory mediators such as NO, prostaglandin E2, and pro-inflammatory cytokines including TNF-α, IL-1β, and IL-6.

For its antidiabetic activity, various extracts—aqueous, methanolic, chloroform, ethyl acetate, acetone, petroleum ester—of dandelion root were tested, highlighting that the root exhibited higher antidiabetic proprieties than leaves and flowers. Also, *Taraxacum*, in combination with other medicinal plants, such as *Momordica charantia*, may demonstrate better antidiabetic activities compared to the individual extracts.

Antioxidant activities of dandelion were exhibited by aqueous extracts and hydro-alcoholic extracts from roots and leaves, which are abundant in ascorbic acid, carotenoids, and polyphenol substances.

The root and leaf of *Taraxacum* were the most commonly tested and may have an antitumoral action that is more significant than the flower of this species. Dandelion extract induces apoptosis through both extrinsic (cell death receptor) and intrinsic (mitochondrial) pathways. Chicoric acid, as a major element of the dandelion extract, present in all components of the plant, induces cell apoptosis in combination treatment with tumor necrosis factor (TNF).

Over the past two decades, studies have highlighted promising applications for Taraxacum extracts in managing chronic inflammatory diseases, oxidative stress-related conditions, cancer prevention and treatment, and digestive disorders.

While the findings from this review are highly promising for the use of dandelion as a medicinal nutraceutical, certain limitations currently exist, including the following: (A) safety concerns include: (1) contamination risk of heavy metals and pesticides, (2) limited data on interactions with conventional medications, (3) allergic reactions like mild skin rashes or other more; (B) regulatory difficulties: (1) chemical composition of dandelion can vary significantly depending on factors like growing conditions, harvesting time, and processing methods, so it is difficult to standardize extracts and guarantee reliable therapeutic effects, (2) there is no universally accepted standard for extracting bioactive compounds from dandelion, (3) authorities in different countries have varying guidelines for evaluating and approving herbal products; (C) scientific knowledge disruptions: (1) incomplete mechanisms of action of the biological active compounds of dandelion, (2) optimal dosage and formulation of dandelion-based treatments, (3) more clinical trials should be made to support its effective use, (4) long-term trials, on realistic interval of time, would offer perceptions of the possible side-effects of diverse extract of the plant; which can complicate the process of bringing dandelion-based treatments in the pharmaceutical perspective and clinical trials.

To address these challenges, further research is needed to develop standardized extraction methods and quality control procedures, investigate the mechanisms of action of dandelion’s bioactive compounds, establish clear regulatory guidelines for herbal medicines, including dandelion-based products conduct rigorous clinical trials to evaluate the safety and efficacy of dandelion-based treatments. By addressing these challenges, we can capitalize on the potential of dandelion as a valuable therapeutic agent and ensure the safety and efficacy of dandelion-based treatments. Therefore, additional research and educational efforts will be required to promote the consumption of this health-promoting multipotent vegetal product.

## Figures and Tables

**Figure 1 ijms-26-00450-f001:**
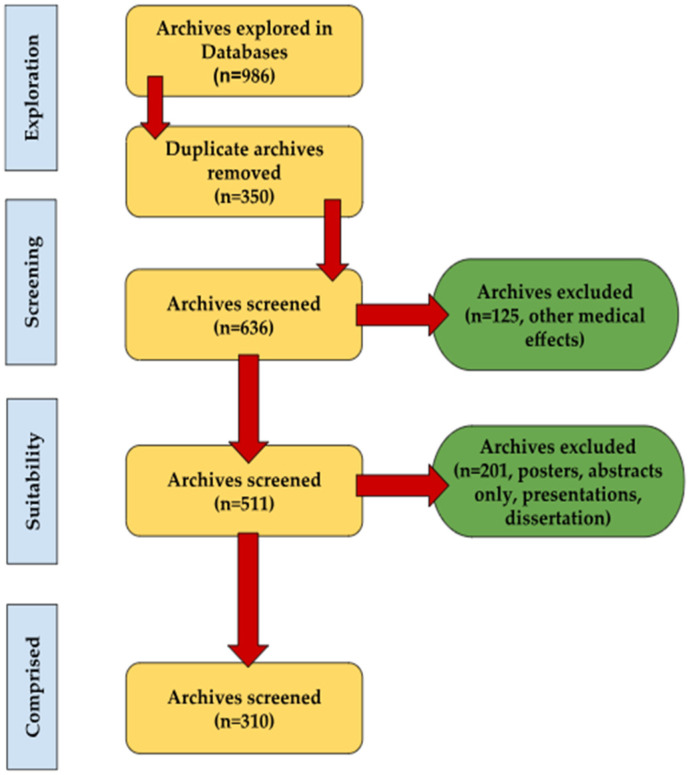
Flow chart describing the selection and screening of literature for systematic reviews using preferred articles.

**Figure 2 ijms-26-00450-f002:**
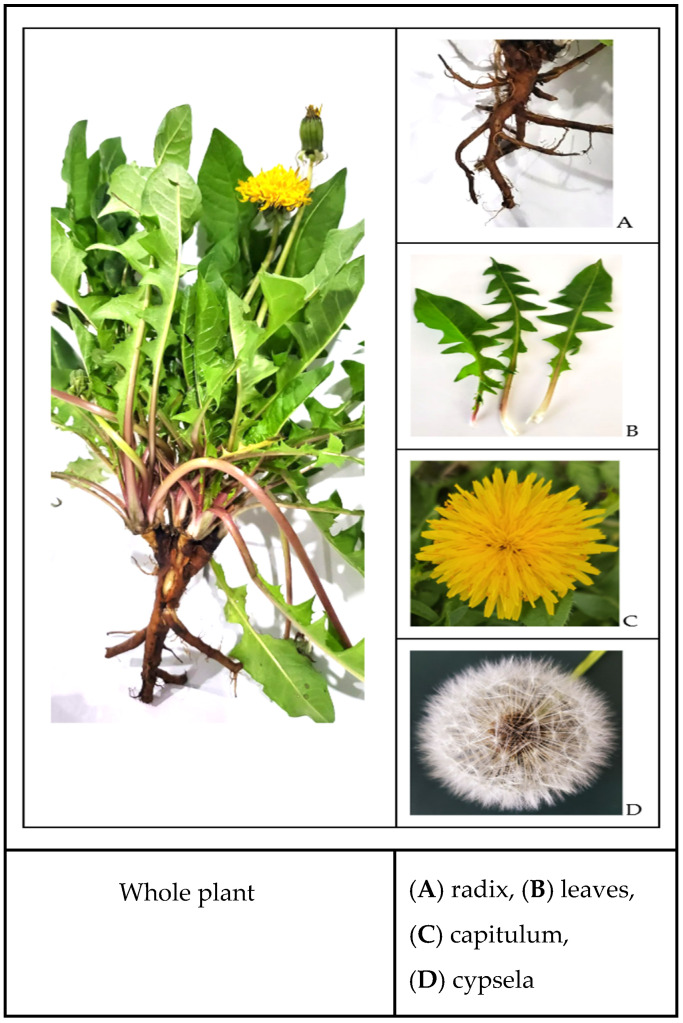
*Taraxacum officinale* L. vegetal organs (original photo).

**Figure 3 ijms-26-00450-f003:**
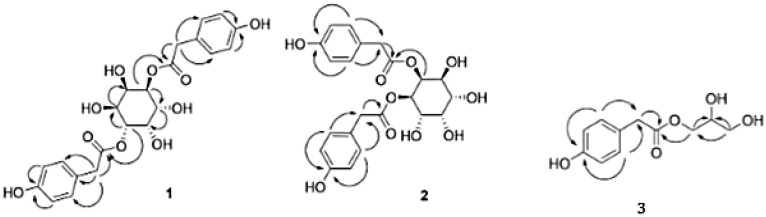
Key HMBC correlations of compounds 1 = taraxinositol A, 2 = taraxinositol B and 3 = taraxinol (adapted after [30]).

**Figure 5 ijms-26-00450-f005:**
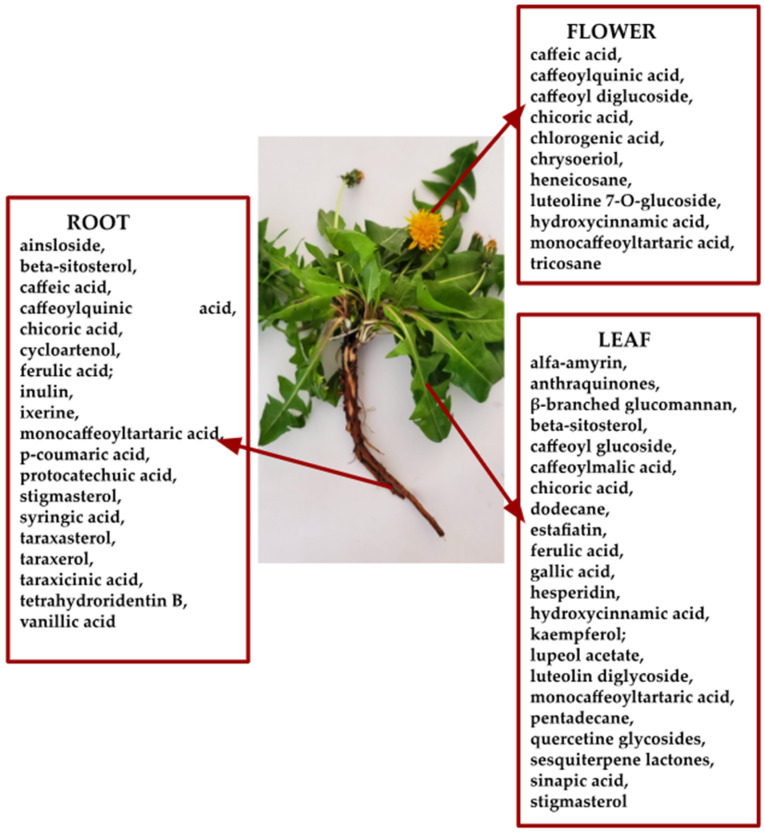
The main active compounds of dandelion (original).

**Figure 7 ijms-26-00450-f007:**
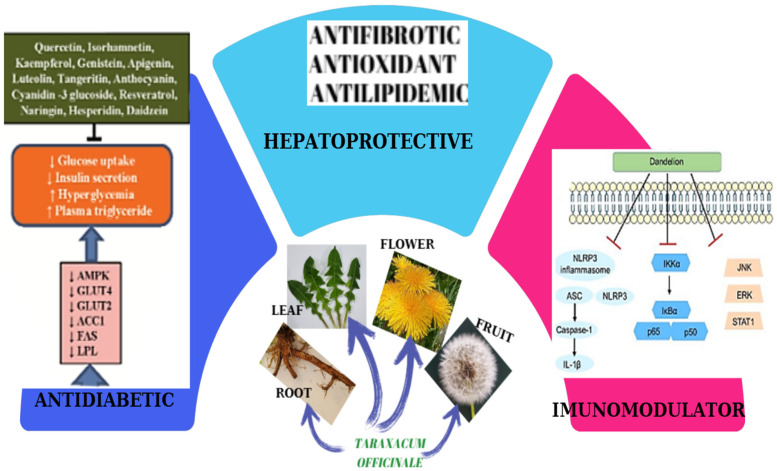
The described antidiabetic, hepatoprotective, and immunomodulator mechanisms of *Taraxacum* extracts (adapted after [6,65,264]).

**Figure 8 ijms-26-00450-f008:**
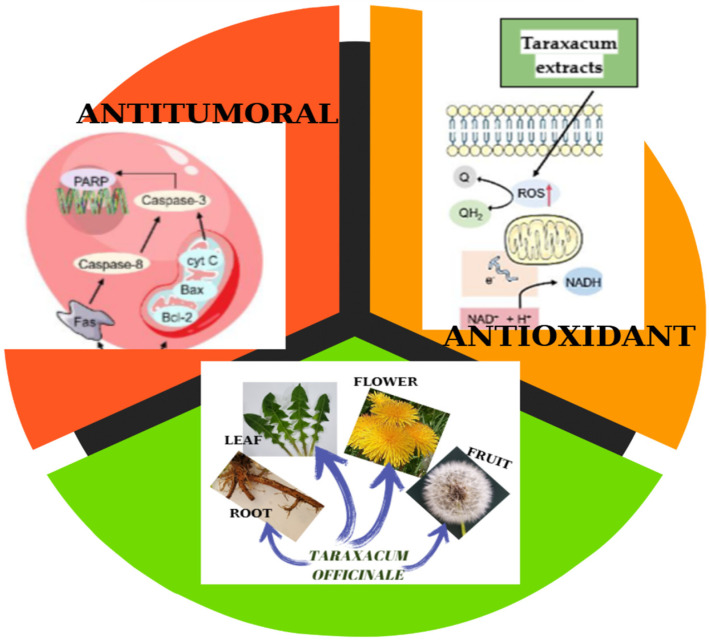
The described antitumoral and antioxidant mechanisms of *Taraxacum* extracts (adapted after [65]).

**Table 1 ijms-26-00450-t001:** The chemical structures of representative compounds named after the *Taraxacum* genus.

Compounds	Name and Structure
Sesquiterpenoids	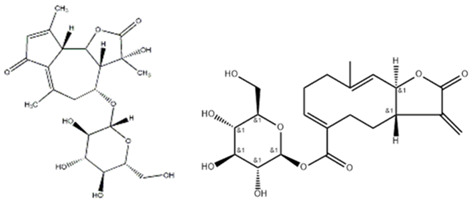 Taraxafolide Taraxinic acid β-D-glucopyranosyl ester
	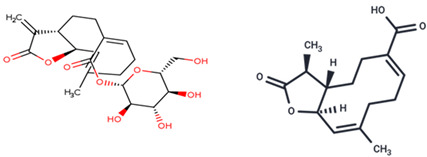 Taraxinic acid β-glucopyranosyl ester 11β,13-Dihydrotaraxinic acid
Phenolics	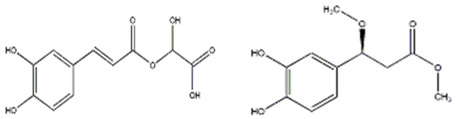 (+)-Taraxafolin B Taraxafolin
Triterpenoids	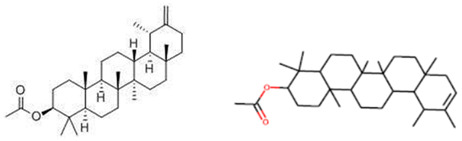 Taraxasteryl acetate Ψ-taraxasteryl acetate
Other types	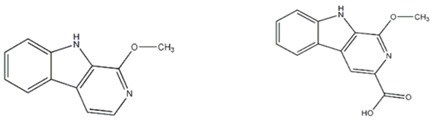 Taraxacine A Taraxacine B
	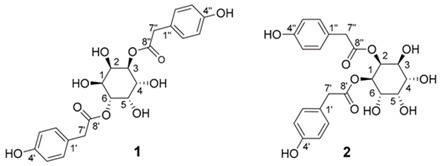 Taraxinositol A Taraxinositol B

**Table 3 ijms-26-00450-t003:** Anti-inflammatory activity of *T. officinale* = *T.o*; *T. mongolicum* = *T.m*; *T. coreanum* = *T.c*.

Vegetal Organ/Phytocompounds	Type of Experiment	Results	Reference
*T. o.* root extract	human neutrophils	MeOH extract inhibits the formation of leukotriene B_4_	[57]
*T. o.* extract	primary cultures of rat astrocytes	treatment of TO (100 and 1000 µg/mL) to astrocytes inhibited the TNF-alpha production by inhibiting Interleukin-1 production	[157]
*T. o.* leaves aqueous extracts	rat mammary microvascular endothelial cells	inhibit both TNF-α and ICAM-1 expression	[158]
*T. o.* dried herbs from Pharmacy	in vivo, octapeptide-induced acute pancreatitis in rats	treatment of TO 10 mg/kg orally administered increased pancreatic levels of HSP60 and HSP72 and decreased the secretion of IL-6 and TNF-α	[159]
*T. o.* dried plant	chicken chorioallantoic assay	ethanol extracts diminished leukocyte levels	[161]
Taraxasterol	BV2 microglia cells culture	dose-dependently inhibited LPS-induced production of TNF-α and IL-1β, suppressed NF-κB activation, and activated the LXRα-ABCA1 signaling pathway	[162]
*T.o.* extract	acute lung injury induced by lipopolysaccharide in mice	oral administration of 2.5, 5, and 10 mg/kg significantly inhibited the inflammatory cytokines TNF-α and IL-6 in the bronchoalveolar lavage fluid	[163]
*T.o.* luteolin, chicoric acid	RAW 264.7 cells	luteolin activates the NF-κB and Akt pathways, while chicoric acid enhances luteolin’s anti-inflammatory activity by attenuating NF-κB activation.	[165]
*T. o.* leaves extract	ovalbumin-sensitized guinea-pig trachea	reduced monocytes, lymphocytes, neutrophils, eosinophils, and basophils	[166]
*T. o.* ethanolic leaf extract	ovalbumin-sensitized guinea-pigs	infiltration of eosinophils and basophils were reduced in the lungs	[167]
*T. o.* taraxasterol	RAW 264.7 macrophages	inhibit mRNA and protein expression levels of iNOS and COX-2	[168]
*T. o.* taraxasterol	murine model of endotoxic shock on mice	reduced TNF-α, IFN-γ, IL-1β, IL-6, NO and PGE_2_ levels in sera	[169]
*T. o.* lipopolysaccharide	RAW 264.7 cells	inhibited NF-κB-mediated inflammation and enhanced Nrf2-mediated antioxidative activity by modulating the PI3K/Akt pathway	[170]
*T. m.* taraxasterol	concanavalin A-induced acute hepatic injury in mice	inhibited pro-inflammatory cytokines TNF-α, IL-6, IL-1β, interferon-γ (IFN-γ), and IL-4	[171]
*T. o.* aerial parts, polyphenolic tincture	rat turpentine-induced inflammation model	Reduced serum oxidative markers: malondialdehyde (MDA), thiols (SH), and nitrites/nitrates (NOx), NF-κB levels	[172]
*T. c.* ethanol extract of aerial parts	lipopolysaccharide-stimulated Caco-2 cells	100 μg/mL extract reduced inducible NO synthase, cyclooxygenase-2, TNF-α, IL-6, and IL-1β	[173]
*T. o.* taraxasterol	drug (APAP-induced liver injury AML12 cells and mice)	promoted Nrf2 and HO-1 expression, suppressed JNK phosphorylation, and decreased the Bax/Bcl-2 ratio and caspase-3 expression	[174]
*T. o.* tincture	rat isoprenaline-induced myocardial infarction models	administration of the tincture led to a decrease in aspartate aminotransferase (AST), alanine aminotransferase (ALT), creatine kinase-MB (CK-MB), and nuclear factor kappa B (NF-κB) levels	[175]

**Table 4 ijms-26-00450-t004:** Antiviral activity of *T. officinale* = *T.o*; *T. farinosum* = *T.f*; *T. mirabile* = *T.m*; *T. campylodes* = *T.c*.

Vegetal Organ/Phytocompounds	Type of Experiment	Results	Reference
*T. f.* and *T. m.* methanol and aqueous extract	in vitro HSV-1/Vero cells system	15.63 μg/mL concentrations of acyclovir showed 100% protection against HSV-1, and 195.31 μg/mL methanol extract of *T. mirabile* only showed 7.08% protection	[181]
*T. o.* chloroform and exane leaf extracts	lavivirus, using the 17D vaccine strain of yellow fever virus as a model	Extract from leaves of 5 months of growth is 8 times more effective than extract from leaves of 2 months of growth	[189]
*T. o.* methanol extract of leaves	human hepatoma (Huh-7) and CHO cell lines transfected with pCR3.1/Flagtag/HCV NS5B gene cloned vector (genotype 1a)	65% inhibition of NS5B expression was documented at nontoxic dose concentration (200 μg/mL)	[190]
*T. o.* methanol extract of leaves	replication of dengue virus serotype 2 in baby hamster kidney BHK-21 cells	extracts at 60 °C showed higher inhibitory effects and present bioactive compounds: luteolin, caffeoylquinic acids, quercetin diclycosides	[191]
*T. o.* water extract of leaves	wild type and mutant forms of SARS-CoV-2 in human HEK293 -hACE2 kidney and A549-hACE2 TMPRSS2 lung cells	Infection of the lung cells was efficiently prevented, and so was virus-triggered pro-inflammatory interleukin 6 secretion	[193]
*T. c.* water extract	viral proteins NS2B/NS3 (DENV-2), NS5B (HCV), and ICP27 (HSV-1)	inhibitory effects of chlorogenic acid (for DENV-2 and HCV), rutin (for HCV and HSV-1), and rosmarinic acid (for DENV-2 and HCV)	[195]

**Table 8 ijms-26-00450-t008:** Immunomodulation activity of *T. officinale* = *T.o*; *T. coreanum* = *T.c*.

Vegetal Organ/Phyto-Compounds	Type of Experiment	Results	Reference
*T. o.* root powder (capsule contained 500 mg)	male albino mice of the CD1 strain and laboratory-raised snails of the species *Biomphalaria alexandrina* were infected with *Schistosoma mansoni miracidia.*	combined treatment with praziquantel and *Taraxacum* root resulted in the most significant improvements—morphological changes in adult worms, reduced worm burden and egg count, decreased granuloma size, altered immune cell distribution, and modulated cytokine levels compared to the infected untreated group	[261]
*T. o.* flower extract	immune response and disease resistance in rainbow trout fed with the extract fed with 3 and 4 g/kg of the extract	dietary supplementation with the extract at 3 g/kg significantly increased total leukocyte and lymphocyte counts, immunoglobulin M, total protein, and lysozyme levels in fish. Additionally, the expression of interleukin-1β and interleukin-6 genes were upregulated in fish fed 3 and 4 g/kg of the extract. The recommended dietary dose is 2.49–2.74 g/kg.	[262]
*T. c.* aerial part chloroform fraction	isolated mouse peritoneal macrophages stimulated in vitro with interferon-γ (IFN-γ) and lipopolysaccharide in a mouse model of lethal septic shock	suppressed the production of nitric oxide (NO) and prostaglandin E2 (PGE2) as inflammatory mediators and repressed the expression of inducible nitric oxide synthase (iNOS) and cyclooxygenase-2 (COX-2); blocked the activation signaling pathways: NF-κB, MAPK, and STAT1	[266]
*T. o.* hot-water and cold-water extract of roots	innate and adaptive immune responses in mice	thioglycollate-induced macrophages cultured with TO-100 and TO-4 produced a significantly higher quantity of various cytokines, such as IL-6 and IL-12	[267]
*T. o.* methanolic extract of roots	HepG2, MCF7, HCT116, and normal Hs27 cell lines	500 µg/mL inhibited the growth of HepG2 cells, and a less pronounced effect on MCF7 and HCT116 cells was registered. No significant effect was observed on Hs27 cells. Also, the treatment improved the phosphorylation of AMPK in HepG2 cells, a key issue in cancer therapy	[268]

**Table 10 ijms-26-00450-t010:** Antioxidant activity of *T. officinale = T.o*.

Vegetal Organ/Phyto-Compounds	Type of Experiment	Results	Reference
Chicoric acid	high-fat diet mice treated with chicoric acid	inhibited the protein expressions of peroxisome proliferator-activated receptor γ (PPARγ) and CCAAT/enhancer-binding protein α (C/EBPα), increasing the activity of antioxidant enzymes and total antioxidant capacity in the liver, regulated the levels of leptin and adiponectin	[230]
*T. o.* methanol fruit extract	estimating antiradical, antiplatelet, and antioxidant properties related to hemostasis	significantly reduced plasma lipid peroxidation and protein thiol oxidation stimulated by H_2_O_2_/Fe at the highest tested concentrations of 10 and 50 µg/mL.	[258]
*T. o.* root	male CD1 strain albino mice and laboratory-bred snails (*Biomphalaria alexandrina*) infected with *S. mansoni* miracidia	potent antioxidant activity at a significant level, suggesting superior scavenging efficacy	[261]
*T. o.* flower water syrup	obese male albino-Wistar rats	increased plasma superoxide dismutase (SOD) activity (1.6-fold) and decreased lipid peroxidation (MDA, 0.81-fold), modulating ACh-induced relaxation but not carbon monoxide-releasing molecule-2 (CORM-2)-induced relaxation in isolated thoracic arteries	[287]
*T. o.* flower ethanol extract	bacterial-lipopolysaccharide-stimulated mouse macrophage RAW264.7 cells	suppressed the stable DPPH radical in a concentration-dependent manner, also exhibited a significant synergistic effect with a-tocopherol in scavenging DPPH radical	[288]
*T. o.* water and ethanol extract	RAW 264.7 cells	both extracts dose-dependently induced HO-1 expression without any cytotoxicity. Water extract activated HO-1 via PI3K/Akt and JNK phosphorylation, while ethanol extract used PI3K/Akt. The antioxidant potential was confirmed by HO-1 inducer (CoPP) and inhibitor (SnPP). Both extracts dose-dependently induced HO-1 expression without cytotoxicity	[297]
*T. o.* water leaves extract	effect of adding T.O extract on the characteristics of raw-ripening pork sausages while reducing the nitrite addition from 150 to 80 mg/kg	dandelion addition (0.5–1%) significantly improved the oxidative stability of nitrite-reduced (80 mg/kg) raw-ripening sausages, as evidenced by ABTS+ and DPPH radical scavenging activity and TBARS assays	[304]

## Data Availability

Data is contained within the article.

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
