# Peer review of "Bioactive Compounds from Vegetal Organs of Taraxacum Species (Dandelion) with Biomedical Applications: A Review"

_ijms, 2025, doi:10.3390/ijms26020450_

Round 1

Reviewer 1 Report

Comments and Suggestions for Authors

This review introduced comprehensive investigations of bioactive compounds from Vegetal Organs of Taraxacum species (Dandelion) with particular emphasis on their biological and pharmacological characteristics. The manuscript is well written and quite organized. However, some concerns should be considered as follows:

1. Abstract: I recommend authors to move lines 21-26 to the intro and replace them with one sentence.

2. Keywords: specific pharmaceutical applications could be added, such as, anti-inflammatory, antimicrobial etc…

3. Intro: the aim of this review should be added at the end or it is reasonable if the authors merged intro with section 2: methodology and strategy.

4. The authors should add a figure using Scopus or Pubmed database, illustrating the No. of publications on this topic during the last 5 years and the advancement of pharmaceutical applications since they discussed these databases in section 2.

5. The extraction process of bioactive compounds or extracts from Taraxacum species should be discussed and add a table concerning the comparison between different methods and the recommended method.

6. Line 172: is this a table or a figure? Please correct. The authors should redraw the chemical structures since I presume they got them as copy and paste with low resolution. Or improve the resolution of images.

7. Fig. 3 should be improved. I would thank the authors for Table 2 since it is informative. The names of bacterial strains, fungi, and plants should be spell out when mentioning for the first time and then abbreviated (the authors mentioned all of them without abbreviations) please check line 602 and 621-622.

8. the authors outlined hepatocellular carcinoma (HCC) (Line 710), please add it to antitumor section.

9. Figure 5: it should be improved and I recommend to separate each biological mechanism and put it under a respective section since the images are extremely minimized.

10. Future perspectives should be discussed, including the limitations of previous reports. 

Reviewer 2 Report

Comments and Suggestions for Authors

The present review article entitled "Bioactive Compounds from Vegetal Organs of Taraxacum species (Dandelion) with Biomedical Applications: A Review" by TAnasa et al  clearly provides a broad overview of phytotherapeutic plants and emphasizes the importance of the Taraxacum genus (dandelion) in traditional and modern medicine. It presents the medicinal applications of dandelion, highlighting its anti-inflammatory, hepatoprotective, and antidiabetic properties. Additionally, the methodology section describes the approach taken to collect scientific literature on Taraxacum’s phytochemistry and pharmacology, mentioning the search strategy, databases, and keywords used. The systematic approach in data collection is clear, with specific inclusion and exclusion criteria for selecting studies.

Moreover, the results discuss the phytochemical profile of Taraxacum, including various bioactive compounds like polyphenols, flavonoids, and terpenes, and the pharmacological activities linked to these (anti-inflammatory, antioxidant, antidiabetic, etc.).

Besides, the discussion section interprets the findings, emphasizing the biomedical potential of Taraxacum and suggesting mechanisms by which its compounds may exert therapeutic effects.

Finally, the conclusion section could be stronger by suggesting specific research directions, like clinical trials or detailed studies on certain bioactive compounds. Also, emphasizing the potential of Taraxacum as a source for new drug candidates would make the conclusion more impactful.

I would like to include some important suggestions to improve the present review manuscript and questions for the authors:

ü A more focused description of Taraxacum’s bioactive compounds and their mechanisms of action could strengthen the introduction, giving readers a better understanding of what makes Taraxacum unique.

ü Including recent clinical studies or ongoing research on Taraxacum's biomedical applications would add more context.

ü Are there studies that compare the efficacy of Taraxacum extracts to standard pharmaceutical treatments for similar conditions?

ü The use of multiple databases strengthens the review by covering a broad range of literature sources.

ü It would improve the section to add more information on how studies were evaluated for quality and relevance. For example, was there any ranking or prioritizing of the data included?

ü A chart or flowchart showing the literature selection process (such as the number of articles reviewed and included) would make the methodology more transparent.

ü Did the authors consider regional differences where Taraxacum grows, as this might affect its chemical composition?

ü Presenting the results in a table could make it easier to read, especially for listing bioactive compounds and their related health effects.

ü Including quantitative data on the concentrations of major compounds (e.g., polyphenols or flavonoids) in different parts of the plant (roots, leaves, flowers) would provide more details for readers.

ü How consistent are the concentrations of these bioactive compounds across different studies? Are there variations due to extraction methods or the age of the plant?

ü Adding a discussion on the limitations of existing studies on Taraxacum, like lack of standardized dosages or limited in vivo research, would give a more balanced view.

ü Mentioning challenges for clinical applications of Taraxacum-based treatments, such as potential regulatory or safety issues, would provide a useful insight.

ü What specific clinical applications do the authors find most promising for Taraxacum extracts?

Reviewer 3 Report

Comments and Suggestions for Authors

The paper is well documented and well written. Unfortunately it is difficult to follow and to apprehend. The information could be made more accessible if appropriate tables and figures are provided summarizing the information. The tables are mandatory - the authors may choose between providing a table for each of the subsections (4.1 - 4.8) in the fourth section or a big table with separate sections corresponding to these 8 subsections. The table(s) should furnish a summary of the experiments described in the article; there should be at least five columns: phytocompounds/taraxacum species studied, type of experiment (cell culture/animal/human) and a brief characterization of cell cultures/animals/human groups used, experimental design, results (effects exerted by the phytocompounds in Taraxacum species), reference.
Ideally, figures should also be provided summarizing the mechanisms demonstrated to be influenced by the various phytocompounds in Taraxacum species.

A few English language errors:
as an herbal remedy - as a herbal remedy
"in the literature data" is not necessary in "characteristics of dandelion vegetal organs in the literature data"

Round 2

Reviewer 1 Report

Comments and Suggestions for Authors

The authors addressed previous comments carefully. However, I observed that the plagiarism percentage of the manuscript is 54% according to the iThenticate analysis provided by MDPI, and I would ask the authors to rephrase these sentences.

Author Response

Dear Mrs./Mr. Reviewer,

We sincerely thank you for the constructive comments and valuable suggestions, which have significantly contributed to improving the quality of our manuscript. 

Comments 1: The authors addressed previous comments carefully. However, I observed that the plagiarism percentage of the manuscript is 54% according to the iThenticate analysis provided by MDPI, and I would ask the authors to rephrase these sentences.

Response 1: Thank you for pointing this out. We agree with this comment. We rephrased in the manuscript, the sentences highlighted in the iThenticate analysis Report provided by MDPI. The final version of our manuscript is attached.

We wish you all the best!

Reviewer 3 Report

Comments and Suggestions for Authors

Do authors have complied with my requests. The article is now suitable for publication.

Author Response

Dear Mrs./Mr. Reviewer,

We sincerely thank you for the constructive comments and valuable suggestions, which have significantly contributed to improving the quality of our manuscript ID: ijms-3325342, for publication in International Journal of Molecular Sciences.

We wish you all the best!